# Task Characteristic and Contrastive Contexts for Improving Generalization in Offline Meta-Reinforcement Learning

## Abstract

Context-based offline meta-reinforcement learning (meta-RL) methods typically extract contexts summarizing task information from historical trajectories to achieve adaptation to unseen target tasks. Nevertheless, previous methods may lack generalization and suffer from ineffective adaptation. Our key insight to counteract this issue is that they fail to capture both task characteristic and task contrastive information when generating contexts. In this work, we propose a framework called task characteristic and contrastive contexts for offline meta-RL (TCMRL), which consists of a task characteristic extractor and a task contrastive loss. More specifically, the task characteristic extractor aims at identifying transitions within a trajectory, that are characteristic of a task, when generating contexts. Meanwhile, the task contrastive loss favors the learning of task information that distinguishes tasks from one another by considering interrelations among transitions of trajectory subsequences. Contexts that include both task characteristic and task contrastive information provide a comprehensive understanding of the tasks themselves and implicit relationships among tasks. Experiments in meta-environments show the superiority of TCMRL over previous offline meta-RL methods in generating more generalizable contexts, and achieving efficient and effective adaptation to unseen target tasks.

## 1 Introduction

Context-based offline meta-reinforcement learning (meta-RL) is an approach for learning how to extract contexts from a series of meta-training tasks and achieving adaptation to new environments. Specifically, contexts encompass crucial statistical information about tasks, which is derived from historical trajectories. Recent methods (Dorfman et al., 2021; Gao et al., 2023; Li et al., 2021b; Wang et al., 2023; Yuan & Lu, 2022) leverage contexts extracted from offline data, instead of extensive online interactions with either real or simulated environments. During the meta-training phase, they learn how to extract contexts from historical trajectories sampled from offline datasets of meta-training tasks. During the meta-testing phase, a few trajectories of unseen target tasks are collected to generate the corresponding contexts. An agent then seeks to adapt efficiently and effectively to these unseen target tasks with extracted contexts.

However, most of these recent context-based offline meta-RL methods face the challenge of *context shift*, where contexts encountered during meta-training and meta-testing may have substantial differences. Context shift happens notably because the behavior policy may overfit the offline datasets during the meta-training phase, leading to mismatches with data of unseen target tasks during the meta-testing phase, and yielding poor performance and generalization. Note that this issue is related to the classical memorization problem in meta-learning (Yin et al., 2020) and the Markov decision process (MDP) ambiguity problem (Li et al., 2020; 2021a).

Our key observation is that the limited generalization of contexts related to previous methods arises from failure to capture *task characteristic information* and *task contrastive information*, both of which are crucial components of task information. Contexts that include both of them provide a comprehensive understanding of each task and implicit relationships among tasks, resulting in improved generalization. First, *task characteristic information* refers to the characteristic of each task, reflecting the consistency of contexts within the same task. Although a particular task corresponds to a series of different historical trajectories, all of them reflect similar task characteristic information. Such information typically arises in transitions related to characteristics of tasks, while these transitions are few in number within a trajectory (Arjona-Medina et al., 2019; Faccio et al., 2022). In contrast, other transitions within the trajectory relate to redundant information, as they commonly occur across most tasks. Our motivation

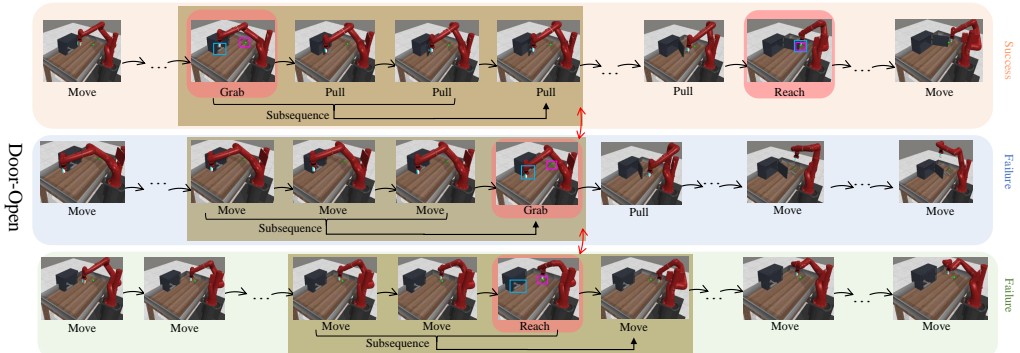

Figure 1: **Motivation of our framework.** In different tasks within the "Door-Open" task set, transitions involving the grabbing and reaching operations relate to task objectives, reflecting the task characteristic information. These transitions vary according to distinct positions of the doorknob and endpoint corresponding to different tasks. Meanwhile, overlooked interrelations among transitions within trajectory subsequences reflects the task contrastive information.

is illustrated in Figure 1. For instance, in the "Door-Open" task set within the Meta-World ML1 environment (Yu et al., 2019), transitions associated with operations about "Grabbing the doorknob" and "Reaching the endpoint" directly relate to the characteristics of tasks while those related to the general moving operations of the robot arm are less important. Moreover, different tasks within the "Door-Open" task set involve distinct characteristic information due to variations in the positions of the doorknob and endpoint. All aforementioned methods fail to effectively identify transitions that are characteristics of tasks and filter out redundant information from general transitions, resulting in a limited understanding of each task and contexts with limited generalization. We aim to identify transitions that are characteristics of a task from the trajectory to capture the task characteristic information and emphasize their roles to improve the generalization of contexts. Second, *task contrastive information* refers to the different task information of various tasks and distinguishes tasks from one another. The trajectories of different tasks comprise transitions related to task dynamics and reward functions, which are core factors of tasks. To generate generalizable contexts, task contrastive information should be extracted from trajectories, highlighting the differences in these factors across tasks. More specifically, such information exists in both the overall trajectory and interrelations among transitions. Previous methods either fail to capture the task contrastive information or only capture it from complete trajectories while overlooking the interrelations among transitions. This leads to an insufficient understanding of implicit relationships among tasks and confusion among contexts of different tasks, hindering the adaptation to unseen target tasks. We aim to discover the overlooked interrelations among transitions for capturing exhaustive task contrastive information that distinguishes tasks from one another.

To this end, we propose a framework called task characteristic and contrastive contexts for offline meta-RL (TCMRL) to improve the generalization of contexts. Specifically, we propose a task characteristic extractor that applies an attention mechanism to identify transitions related to characteristics of tasks and assign high attention weights to them when generating contexts for capturing task characteristic information. To effectively optimize the task characteristic extractor, we introduce a context-based reward estimator and design specific loss functions from the perspectives of positive and negative reward estimation, and sparsity in attention weights. Additionally, we propose a task contrastive loss to discover the overlooked interrelations among transitions from trajectory subsequences. Moreover, the extracted interrelations are extended to the entire trajectory with these subsequences as basic units for capturing exhaustive task contrastive information. In summary, TCMRL improves the generalization of contexts by capturing comprehensive task information that includes both *task characteristic information* and *task contrastive information*, enabling more efficient and effective adaptation to unseen target tasks. The main contributions of TCMRL are fourfold:

- We experimentally demonstrate that the issue of context shift arises from a lack of both task characteristic information and task contrastive information, and capture them from trajectories separately to improve the generalization in offline meta-RL.
- We propose a task characteristic extractor to identify and emphasize transitions related to task characteristics, and introduce a context-based reward estimator and a series of specific loss functions for optimization.

- We propose a task contrastive loss that favors the learning of task information that distinguishes tasks from one another by discovering overlooked interrelations among transitions from trajectory subsequences.
- We demonstrate the effectiveness of the proposed TCMRL through extensive experiments on the MuJoCo environments and the Meta-World ML1 task sets, and results show significant performance improvements compared with previous meta-RL methods.

## 2 RELATED WORK

**Meta-reinforcement learning.** Meta-reinforcement learning aims to acquire learning strategies from a series of meta-training tasks and achieve adaptation to unseen target meta-testing tasks. Previous meta-RL studies can be primarily categorized into two distinct methods: context-based methods and optimization-based methods. Context-based methods encode contexts from the critical statistical information about tasks, which is generally presented in the form of history trajectories. This process is commonly accompanied by the utilization of recurrent (Fakoor et al., 2020; Wang et al., 2017), recursive (Mishra et al., 2018), or probabilistic (Rakelly et al., 2019; Zintgraf et al., 2020) structures. Moreover, optimization-based methods (Finn et al., 2017; Foerster et al., 2018; Houthooft et al., 2018) formalize the process of the task adaptation as the execution of policy gradients over limited samples, aiming to acquire an optimal initialization of the policy. TCMRL is most closely related to the context-based meta-RL.

**Context-based offline meta-reinforcement learning.** Context-based offline meta-RL methods focus on acquiring generalizable contexts from offline datasets of historical trajectories, rather than relying on online interactions with environments during the meta-training phase. It aims to adapt to unseen target tasks during the meta-testing phase. MBML (Li et al., 2020) and BOReL (Dorfman et al., 2021) assume the prior knowledge of reward functions across diverse tasks. FOCAL (Li et al., 2021b) utilizes behavior regularization to restrict the task inference while FOCAL++ (Li et al., 2021a) enhances it by introducing attention mechanisms and contrastive learning. SMAC (Pong et al., 2022) employs semi-supervised learning that introduces additional online data but heavily relies on annotation functions extracted from offline datasets. CORRO (Yuan & Lu, 2022) improves the generalization of contexts through contrastive learning. IDAQ (Wang et al., 2023) leverages a return-based uncertainty quantification to ensure in-distribution contexts of tasks. CSRO (Gao et al., 2023) designs a max-min mutual information representation learning mechanism to reduce the impact of context shift. However, FOCAL, SMAC, IDAQ and CSRO rely on the mean context encoding that treats each transition within a trajectory individually and assigns them the same attention weights, failing to identify transitions related to task characteristics and learn task information that distinguishes tasks from one another. FOCAL++ and CORRO replace the mean context encoding with attention mechanisms but lack focused optimization. They only capture coarse task characteristic information and a portion of the task contrastive information, overlooking the interrelations among transitions. In contrast to these existing studies, TCMRL focuses on capturing task characteristic information and task contrastive information to improve the generalization of contexts, leading to efficient and effective adaptation to unseen target tasks.

## 3 PRELIMINARIES

The formulation of a reinforcement learning (RL) task commonly takes the form of a fully observable Markov decision process (MDP), which can be defined as a tuple $M = \langle \mathcal{S}, \mathcal{A}, p, r, \gamma, \rho_0 \rangle$. $\mathcal{S}$ is the state space, $\mathcal{A}$ is the action space, $s \in \mathcal{S}$ and $a \in \mathcal{A}$ respectively represent the state and action at time-step $t$, $p(s^{t+1}|s^t, a^t)$ is the transition dynamics, $r(s^t, a^t)$ is the reward function, $\rho_0$ is the initial state distribution, and $\gamma \in [0, 1)$ is the discount factor for future rewards. A stochastic policy is a distribution $\pi(a_i^t|s_i^t)$ of actions. Nevertheless, context-based offline meta-RL is generally formalized as partially observable Markov decision processes (POMDPs) (Kaelbling et al., 1998), where states obtained from environments remain only partially visible. It assumes that the information of each task is the unobservable part called the context and the agent needs to collect it from offline data as one of the conditions to make decisions: $a_i^t \sim \pi(a_i^t|s_i^t, c_i)$, where $c_i$ is the context related to the task information of task $\mathcal{T}_i$ and the complete state is formed by combining $s_i^t$ and $c_i$. Moreover, the definition of the marginal state distribution at time-step $t$ is $\mu_\pi^t(s_i^t)$ and the primary goal of the agent, which is the same in both MDPs and POMDPs, is to maximize the objective function $max_\pi \mathcal{J}_{\mathcal{M}}(\pi) = \mathbb{E}_{s_i^t \sim \mu_\pi^t, a_i^t \sim \pi}[\sum_{t=0}^{\infty} \gamma^t r(s_i^t, a_i^t)]$, which represents the expectation of the accumulated rewards over time.

As a context-based offline meta-RL method, TCMRL assumes access to a set of $n_{task}$ meta-training tasks $\mathbb{T} = \{\mathcal{T}_1, ..., \mathcal{T}_{n_{task}}\}$ and a set of unseen target tasks $\mathbb{T}^*$. Each task is individually modeled as a

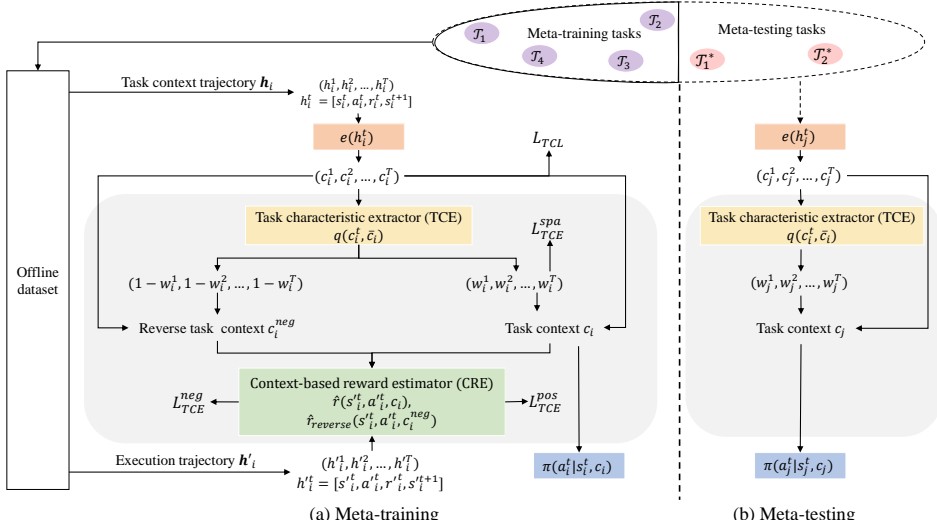

Figure 2: **Framework overview.** **(a) Meta-training** meta-trains a context encoder $e(h_i^t)$, a task characteristic extractor $q(c_i^t, \bar{c}_i)$, a context-based reward estimator $\hat{r}(s_i^t, a_i^t, c_i)$ and a context-based policy $\pi(a_i^t|s_i^t, c_i)$. The context-based reward estimator $\hat{r}(s_i^t, a_i^t, c_i)$ is used to optimize $q(c_i^t, \bar{c}_i)$ with $L_{TCE}^{spa}$, $L_{TCE}^{pos}$ and $L_{TCE}^{neg}$. Task contrastive loss $L_{TCL}$ discovers interrelations among transitions. **(b) Meta-testing** utilizes the meta-trained modules $e(h_i^t)$, $q(c_i^t, \bar{c}_i)$ and $\pi(a_i^t|s_i^t, c_i)$ for efficient and effective adaptation to unseen target tasks with contexts extracted from a few trajectories collected from them.

POMDP. The set of offline datasets $\mathbb{D} = \{\mathcal{D}_1, ..., \mathcal{D}_{n_{task}}\}$ corresponds to the set of meta-training tasks. More details about preliminaries of context-based offline meta-RL can be found in Appendices B and C.

## 4 METHOD

As illustrated in Figure 2, TCMRL consists of two main phases: meta-training and meta-testing. Specifically, during the meta-training phase, TCMRL learns how to extract contexts $c_i$ from historical trajectories $h_i$ sampled from the offline dataset $\mathcal{D}_i$, which corresponds to the meta-training task $\mathcal{T}_i$. During the meta-testing phase, trajectories $h_j$ corresponding to the unseen target task $\mathcal{T}_j$ are collected to extract the contexts $c_j$. TCMRL then utilizes $c_j$ to achieve efficient and effective adaptation to $\mathcal{T}_j$.

### 4.1 META-TRAINING

TCMRL operates in the meta-training phase to capture task characteristic information and task contrastive information separately. It (1) applies our task characteristic extractor (TCE) to identify and emphasize transitions related to the task characteristic within the trajectory and optimizing it through our context-based reward estimator from three perspectives, hence capturing the task characteristic information; and (2) constructs task contrastive loss to discover the overlooked interrelations among transitions from trajectory subsequences, for capturing task contrastive information.

#### 4.1.1 TASK CHARACTERISTIC EXTRACTOR

**Structure.** Previous context-based offline meta-RL methods (Gao et al., 2023; Li et al., 2021b; Wang et al., 2023) typically employ a context encoder $e(h_i^t)$ to encode each transition $h_i^t$ within the historical trajectory $h_i$ into the representation $c_i^t$, where $h_i$ corresponds to the meta-training task $\mathcal{T}_i$ and consists of $T$ time-steps. Each transition $h_i^t = (s_i^t, a_i^t, r_i^t, s_i^{t+1})$ is composed of state $s_i^t$, action $a_i^t$, reward $r_i^t$ and next state $s_i^{t+1}$. Subsequently, these methods treat each representation $c_i^t \in \{c_i^t\}_{t=1}^T$ with equal weight and aggregate them through $\bar{c}_i = mean(\{c_i^t\}_{t=1}^T)$.

For a trajectory $h_i$, which is sampled from the offline dataset $\mathcal{D}_i$ of task $\mathcal{T}_i$ and composed of transitions $\{h_i^t\}_{t=1}^T$, we regard $\bar{c}_i$ as a coarse context that indeed obtains partial task information. This is because it overemphasizes redundant information from less important transitions, rather than identifying and

emphasizing transitions related to the task characteristics. Then, we propose a task characteristic extractor $q(c_i^t, \bar{c}_i)$ that assigns importance scores $(score_i^t)_{t=1}^T$ for all transition representations $\{c_i^t\}_{t=1}^T$ based on $\bar{c}_i$. It aims to identify transitions, within the trajectory $\boldsymbol{h}_i$, that are task characteristic of the task $\mathcal{T}_i$ and assign them high scores for capturing the task characteristic information. In contrast, most general transitions within the trajectory $\boldsymbol{h}_i$ are assigned low importance scores since they appear across many tasks and are associated with redundant information. To aggregate transition representations $\{c_i^t\}_{t=1}^T$ into the context $c_i$ that includes task characteristic information based on their scores $\{(score_i^t)\}_{t=1}^T$, we generate a sequence of attention weights $\boldsymbol{w}_i = \{(w_i^t)\}_{t=1}^T$ with a softmax function, where $w_i^t \in [0,1]$, and $\sum_{t=1}^T w_i^t = 1$. The complete process is as follows:

$$c_i^t = e(h_i^t), \tag{1}$$

$$\bar{c}_i = mean(c_i^1, c_i^2, ..., c_i^T) \tag{2}$$

$$score_i^t = q(c_i^t, \bar{c}_i), \tag{3}$$

$$(w_i^1, ..., w_i^T) = softmax(score_i^1, ..., score_i^T), \tag{4}$$

$$c_i = \sum_{t=1}^T w_i^t \cdot c_i^t. \tag{5}$$

**Optimization.** Inspired by the proposition of Liu et al. (2023) to detect critical frames in videos, we design specific loss functions to optimize the task characteristic extractor from three distinct perspectives for effectively capturing the task characteristic information. Initially, we optimize the task characteristic extractor from the perspective of sparsity corresponding to the sequence of attention weights $(w_i^t)_{t=1}^T$. Our objective is to accurately assign higher importance scores and corresponding attention weights to these transitions that are characteristic of the task, as only a few key transitions within the trajectory provide the main task characteristic information (Arjona-Medina et al., 2019; Faccio et al., 2022). Furthermore, because of the properties of attention weights ($\sum_{t=1}^T w_i^t = 1$), assigning high attention to general transitions results in that contexts include more redundant information instead of task characteristic information. Therefore, since it is difficult to constrain the importance scores $(score_i^t)_{t=1}^T$, we impose a strict requirement on the overall sparsity of the sequence of attention weights $(w_i^t)_{t=1}^T$. This constraint serves to mitigate the risk of excessive weight allocation to general transitions for effectively capturing the task characteristic information. The learning objective of sparsity in attention weights $L_{TCE}^{spa}$ is implemented through the $L_1$ regularization as follows:

$$L_{TCE}^{spa} = \sum_{t=1}^T \|\boldsymbol{w}_i\|_1. \tag{6}$$

To better optimize our task characteristic extractor, a way to measure how well the context includes task characteristic information is essential. We reformulate a context-based reward estimator $\hat{r}(s_i^t, a_i^t, c_i)$, which is different from the conventional reward estimator $\hat{r}(s_i^t, a_i^t)$ widely used in RL. In $\hat{r}(s_i^t, a_i^t)$, reward estimation is confined to the current state $s_i^t$ and the action $a_i^t$. In contrast, $\hat{r}(s_i^t, a_i^t, c_i)$ incorporates the context $c_i$ as an additional input to provide task information. We employ it at the level of transitions within a trajectory. For every input state, action and context, $\hat{r}(s_i^t, a_i^t, c_i)$ performs the reward estimation and contexts that effectively represent task information lead to accurate estimations. More details of the context-based reward estimator can be found in Appendix F.3.

The remaining two perspectives encompass the positive and negative reward estimation obtained through supervised learning with the context-based reward estimator. Additionally, these two perspectives utilize the context, which is jointly generated by the context encoder $e(h_i^t)$ and the task characteristic extractor $q(c_i^t, \bar{c}_i)$, as one of the inputs to the context-based reward estimator $\hat{r}(s'_i^t, a'_i^t, c_i)$. The additional inputs come from another trajectory $\boldsymbol{h}'_i = \{h'_i^t\}_{t=1}^T$, which is called the execution trajectory and sampled from the offline dataset $\mathcal{D}_i$ related to the same task $\mathcal{T}_i$. Each transition $h'_i^t = (s'_i^t, a'_i^t, r'_i^t, s'_i^{t+1})$ within $\boldsymbol{h}'_i$ has the same components as $h_i^t$ in $\boldsymbol{h}_i$.

We design $L_{TCE}^{pos}$ to optimize the task characteristic extractor from the perspective of positive reward estimation for assigning high attention weights to transitions, within a trajectory, that are the task characteristic of a task and capturing the task characteristic information. Specifically, a context $c_i$ of the task $\mathcal{T}_i$ is generated with the task characteristic extractor, which identifies and emphasizes transitions that are the task characteristic of $\mathcal{T}_i$. Therefore, if the context-based reward estimator can make accurate predictions for transitions within $\boldsymbol{h}'_i$ under the condition of $c_i$, it indicates that the task characteristic extractor effectively captures task characteristic information from $\boldsymbol{h}_i$. The

objective of $L_{TCE}^{pos}$ is to optimize the task characteristic extractor by minimizing estimation errors of the context-based reward estimator under the condition of $c_i$. The learning objective is as follows:

$$L_{TCE}^{pos} = \sum_{t=1}^{T} (\hat{r}(s'^t_i, a'^t_i, c_i) - r'^t_i)^2. \tag{7}$$

Meanwhile, $L_{TCE}^{neg}$ is designed to optimize the task characteristic extractor from another perspective of negative reward estimation. Specifically, while the task characteristic extractor assigns transitions related to the task characteristic with greater importance scores and thus higher attention weights, it simultaneously reduces the attention weights of the remaining transitions. This distinction is reflected in the importance scores $(score_i^t)_{t=1}^T$ and the subsequent attention weights $(w_i^t)_{t=1}^T$. Consequently, an additional sequence of negative weights $(1 - w_i^t)_{t=1}^T$ is generated, and through a similar process, the sequence of transition representations $\{c_i^t\}_{t=1}^T$ is aggregated to the reverse context $c_i^{neg}$ ($c_i^{neg} = \sum_{t=1}^T (1 - w_i^t) \cdot c_i^t$). Notably, the sum of $(1 - w_i^t)_{t=1}^T$ is not 1, but this does not affect the calculation process. In this setup, the less important transitions play the more important roles, causing the reverse context $c_i^{neg}$ to primarily capture redundant information from general transitions within the trajectory, rather than the task characteristic information. Therefore, we design $L_{TCE}^{neg}$ to mitigate the impact of redundant information by preventing the context-based reward estimator from making accurate reward predictions when $c_i^{neg}$ is applied as a condition. To achieve this, we construct negative rewards for each transition $h'^t_i$ via adding random noise to the reward $r'^t_i$, serving as the corresponding incorrect estimation targets. In detail, we define $r'^{t\,neg}_i = r'^t_i + r^{noise}$, where $r^{noise}$ is sampled from a Gaussian distribution of noise. Instead of directly designing $L_{TCE}^{neg}$ around incorrect reward estimation, we induce reward estimation conditioned on $c_i^{neg}$ to approximate the corresponding negative rewards. It allows us to design both $L_{TCE}^{pos}$ and $L_{TCE}^{neg}$ with a similar structure. The learning objective is as follows:

$$L_{TCE}^{neg} = \sum_{t=1}^{T} (\hat{r}_{reverse}(s'^t_i, a'^t_i, c^{neg}_i) - r'^{t\,neg}_i)^2. \tag{8}$$

More experimental analysis about the negative reward $r'^{t\,neg}_i$ can be found in Appendix G.3.

Notably, the context encoder, task characteristic extractor and context-based reward estimator are implemented with neural networks and trained simultaneously within TCMRL, without any sequential dependencies. We employ $L_{TCE}^{pos}$ (Eq. 7) to train all of them, while $L_{TCE}^{neg}$ (Eq. 8) is not used to optimize the context-based reward estimator. This is because $L_{TCE}^{neg}$ is associated with the bias of reward estimation and the negative reward $r'^{t\,neg}_i$ is not an exact or accurate estimation target. Overall, we simultaneously minimize all three losses $L_{TCE}^{spa}$, $L_{TCE}^{pos}$ and $L_{TCE}^{neg}$, to optimize the task characteristic extractor for capturing the task characteristic information.

### 4.1.2 Task contrastive loss

Task contrastive information is essential for improving the generalization of contexts as it reflects the differences in task dynamics and reward functions across tasks. TACO (Zheng et al., 2023) is a method that learns state and action representations related to the task dynamics by maximizing the mutual information between current states paired with action sequences and representations of the future states. We generalize this idea into a mutual information objective to capture the structural information of subsequences and propose a task contrastive loss that discovers the overlooked interrelations among transitions from trajectory subsequences to capture task contrastive information that distinguishes tasks from one another. Furthermore, we extend the interrelations among transitions to the entire trajectory with these subsequences as basic units for obtaining exhaustive task contrastive information. Such information leads to a complete understanding of the implicit relationships among tasks. To the best of our knowledge, we are the first to discover overlooked interrelations among transitions from trajectory subsequences through contrastive learning for capturing exhaustive task contrastive information.

As shown in Figure 3, for a subsequence of length $K$ corresponding to the task $\mathcal{T}_i$, we regard the average of transition representations from the first $K - 1$ steps as prior context representation and consider the $K$-th step transition representation as target context representation. Then, we maximize the mutual information $I$ between the prior context representation and the target transition representation:

$$I(m_i^t; c_i^{t+K-1}), \tag{9}$$

where $K$ is a fixed hyperparameter that satisfies $K > 1$ and $m_i^t$ is a convenient representation for $mean(c_i^t, ..., c_i^{t+K-2})$. We approximate the lower bound of the mutual information with the InfoNCE

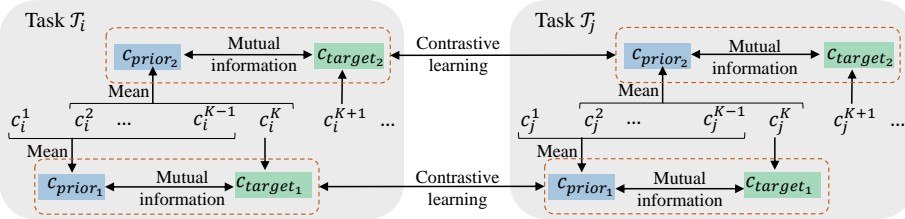

Figure 3: **Computation process of task contrastive loss.** TCMRL discovers interrelations among transitions from a complete trajectory by using subsequences of length $K$ as the fundamental units. Such interrelations among transitions are constructed based on the mutual information between the prior and target transition representations and used to capture task contrastive information.

loss function (van den Oord et al., 2018), which is defined as follows:

$$L_{InfoNCE} = -\log \frac{\exp(z^q \cdot z_+^k / \tau)}{\sum_{i=1}^{B} \exp(z^q \cdot z_i^k / \tau)}, \qquad (10)$$

where $\tau$ is the temperature hyperparameter, $z^q$ is a query vector and $\{z_1^k, ..., z_B^k\}$ is a set of $B$ key vectors. We assume the key $z_+^k \in \{z_1^k, ..., z_B^k\}$ is the only one matching $z^q$.

To construct the task contrastive loss, we operate in sequences of transition representations $\{\{c_i^t\}_{t=1}^{T}\}_{i=1}^{B}$ related to $B$ tasks $\{\mathcal{T}_i\}_{i=1}^{B}$ through two distinct levels of steps. First, we compute the mutual information $I$ in Eq. 9 with Eq. 10 to discover interrelations among transitions in subsequences $\{(c_i^t, c_i^{t+1}, ..., c_i^{t+K-1})\}_{i=1}^{B}$. It relies on the matching relationship between the prior and target context representations within the subsequence of the same task, as both reflect the same task dynamic and reward function, and share a sequential relationship within the subsequence. However, this process only considers the interrelations between each transition within $m_i^t$ and the transition corresponding to $c_i^{t+K-1}$. Second, we extend these interrelations to entire trajectories with sets of subsequences as the basic units. This operation further discovers interrelations among each transition and up to $K$ surrounding transitions while except for the first and last $K-1$ transitions in the trajectory, all other transitions contribute to both $m_i^t$ and $c_i^{t+K-1}$. The complete computation process is as follows:

$$L_{TCL} = -\frac{1}{T-K+1} \frac{1}{B} \sum_{t=1}^{T-K+1} \sum_{i=1}^{B} \log \frac{m_i^t \mathcal{W} c_i^{t+K-1}}{\sum_{l=1}^{B} m_l^t \mathcal{W} c_l^{t+K-1}}, \qquad (11)$$

where $\mathcal{W}$ is a parameter of the weight, which is learnable and provides a similarity measure between $m_i^t$ and $c_i^{t+K-1}$. Although the computation of $L_{TCL}$ in Eq. 11 appears to involve a double loop with time complexity related to both $B$ and $T$, it can be computed through matrix operations with time complexity of the inner loop. The inner loop can be written in a matrix-form as follows:

$$\mathcal{L}_{inner} = -\sum_{i=1}^{B} \log \frac{m_i^t \mathcal{W} c_i^{t+K-1}}{\sum_{l=1}^{B} m_l^t \mathcal{W} c_l^{t+K-1}} = -\text{Tr}(M), \quad M_{ij} = \log \frac{m_i^t \mathcal{W} c_i^{t+K-1}}{\sum_{l=1}^{B} m_l^t \mathcal{W} c_l^{t+K-1}}. \qquad (12)$$

Meanwhile, the outer loop primarily relates to the parallel computations of prior context representations. In summary, our task contrastive loss can discover overlooked interrelations among transitions and capture exhaustive task contrastive information, leading to contexts with generalization.

### 4.2 META-TESTING

During the meta-testing phase, TCMRL aims to achieve efficient and effective adaptation to the set of unseen target tasks $\mathbb{T}^*$ with our trained context encoder $e(h_j^t)$, task characteristic extractor $q(c_j^t, \bar{c}_j)$ and context-based policy $\pi(a_j^t | s_j^t, c_j)$. When facing an unseen target task $\mathcal{T}_j$, TCMRL begins by collecting a limited number of trajectories $\boldsymbol{h}_j = \{h_j^t\}_{t=1}^{T}$. Subsequently, we utilize $e(h_j^t)$ to generate representations $\{c_j^t\}_{t=1}^{T}$ for transitions $\{h_j^t\}_{t=1}^{T}$. Then, $q(c_j^t, \bar{c}_j)$ inputs $c_j^t$ and $\bar{c}_j$ and outputs the importance score $score_j^t$. Furthermore, the sequence of scores $(score_j^t)_{t=1}^{T}$ generated by Eq. 3 is transformed into the sequence of attention weights $(w_j^t)_{t=1}^{T}$ with Eq. 4. Finally, by the aggregation of $\{c_j^t\}_{t=1}^{T}$ and $(w_j^t)_{t=1}^{T}$ with Eq. 5, we obtain the context $c_j$, which is a partial input to $\pi(a_j^t | s_j^t, c_j)$ for generating actions.

Table 1: Performance in meta-environments with normalized scores.

| Task set/Environment | TCMRL (ours) | IDAQ | CSRO | CORRO | FOCAL++ | FOCAL | MACAW | BOReL |
|---|---|---|---|---|---|---|---|---|
| Basketball | **0.82±0.11** | 0.64±0.15 | 0.58±0.10 | 0.57±0.04 | 0.71±0.25 | 0.41±0.24 | 0.00±0.00 | 0.00±0.00 |
| Box-Close | **0.62±0.09** | 0.51±0.11 | 0.51±0.02 | 0.60±0.03 | 0.44±0.03 | 0.15±0.09 | 0.36±0.11 | 0.05±0.01 |
| Button-Press-Topdown | **0.81±0.12** | 0.57±0.11 | 0.66±0.09 | 0.55±0.14 | 0.51±0.10 | 0.45±0.10 | 0.38±0.36 | 0.02±0.02 |
| Dial-Turn | **0.98±0.01** | 0.91±0.05 | 0.81±0.09 | 0.87±0.07 | 0.80±0.13 | 0.84±0.09 | 0.00±0.00 | 0.00±0.00 |
| Disassemble | **0.59±0.13** | 0.41±0.14 | 0.56±0.06 | 0.49±0.06 | 0.32±0.08 | 0.25±0.04 | 0.05±0.00 | 0.04±0.00 |
| Door-Close | **1.01±0.00** | 0.99±0.00 | 0.74±0.18 | 0.98±0.01 | 1.01±0.00 | 0.97±0.01 | 0.00±0.00 | 0.37±0.19 |
| Door-Lock | **0.99±0.00** | 0.97±0.01 | 0.94±0.02 | 0.89±0.05 | 0.96±0.00 | 0.90±0.02 | 0.25±0.11 | 0.14±0.00 |
| Door-Unlock | **1.18±0.02** | 1.11±0.02 | 1.13±0.01 | 1.15±0.01 | 1.11±0.02 | 0.97±0.03 | 0.11±0.01 | 0.13±0.03 |
| Door-Open | **1.00±0.00** | 0.94±0.02 | 0.98±0.00 | 0.91±0.05 | 0.92±0.01 | 0.76±0.13 | 0.06±0.01 | 0.11±0.01 |
| Drawer-Close | **1.01±0.01** | 0.99±0.02 | 1.00±0.01 | 0.94±0.02 | 0.97±0.01 | 0.96±0.04 | 0.53±0.50 | 0.00±0.00 |
| Drawer-Open | **0.90±0.03** | 0.82±0.06 | 0.54±0.21 | 0.74±0.04 | 0.84±0.05 | 0.64±0.10 | 0.11±0.02 | 0.10±0.00 |
| Faucet-Open | **1.08±0.02** | 1.05±0.02 | 1.05±0.01 | 1.07±0.00 | 1.06±0.00 | 1.01±0.02 | 0.08±0.04 | 0.12±0.05 |
| Hand-Insert | **0.72±0.05** | 0.63±0.04 | 0.64±0.02 | 0.63±0.13 | 0.56±0.06 | 0.29±0.07 | 0.02±0.01 | 0.00±0.00 |
| Lever-Pull | **0.86±0.02** | 0.84±0.02 | 0.79±0.03 | 0.81±0.03 | 0.62±0.06 | 0.72±0.07 | 0.20±0.16 | 0.05±0.00 |
| Peg-Insert-Side | **0.45±0.05** | 0.30±0.04 | 0.27±0.14 | 0.36±0.10 | 0.19±0.07 | 0.08±0.03 | 0.00±0.00 | 0.00±0.00 |
| Pick-Out-Of-Hole | **0.71±0.06** | 0.25±0.25 | 0.54±0.13 | 0.52±0.14 | 0.29±0.17 | 0.15±0.16 | 0.59±0.06 | 0.00±0.00 |
| Pick-Place | **0.32±0.09** | 0.19±0.03 | 0.11±0.03 | 0.25±0.05 | 0.14±0.03 | 0.07±0.02 | 0.05±0.05 | 0.00±0.00 |
| Reach | **0.92±0.03** | 0.85±0.03 | 0.75±0.20 | 0.43±0.36 | 0.87±0.04 | 0.62±0.05 | 0.63±0.04 | 0.04±0.01 |
| Soccer | **0.60±0.06** | 0.44±0.04 | 0.54±0.11 | 0.58±0.07 | 0.29±0.03 | 0.11±0.03 | 0.38±0.31 | 0.04±0.02 |
| Window-Close | **0.95±0.01** | 0.93±0.01 | 0.93±0.02 | 0.92±0.01 | 0.94±0.01 | 0.79±0.01 | 0.54±0.44 | 0.03±0.00 |
| Sparse-Point-Robot | **12.98±0.29** | 7.74±0.68 | – | – | 11.59±0.15 | 11.66±0.46 | 0.00±0.00 | 0.00±0.00 |
| Half-Cheetanh-Vel | **-79.7±11.3** | -133.4±23.9 | -114.5±14.0 | -113.2±17.2 | -116.7±14.9 | -117.7±13.6 | -234.0±23.5 | -301.4±36.8 |
| Point-Robot-Wind | **-4.75±0.26** | -6.03±0.22 | – | – | -4.89±0.31 | -5.46±0.26 | – | – |
| Hopper-Rand-Params | **368.62±10.37** | 325.74±27.09 | 331.65±33.62 | 293.32±17.49 | 318.86±20.14 | 314.41±29.00 | – | 52.82 |
| Walker-Rand-Params | **354.97±19.72** | 324.04±31.40 | 316.81±16.31 | 301.49±5.06 | 313.02±24.22 | 303.07±4.28 | 311.68 | 269.74 |

Notably, the data collection process can be divided into two distinct stages. In the initial stage, the agent randomly samples actions $a_j^t$ to collect the trajectory $\boldsymbol{h}_j$ for extracting context $c_j$, while in the subsequent stage, actions $a_j^t$ are sampled based on the context-based policy $\pi(a_j^t|s_j^t, c_j)$.

Pseudo-codes of both the meta-training and meta-testing phases can be found in Appendix A.

## 5 EXPERIMENTS

We evaluate TCMRL on two main issues: (1) whether generalizable contexts can be extracted and (2) whether an efficient and effective adaptation to unseen target tasks can be achieved. Our code is available at `https://anonymous.4open.science/r/TCMRL/`.

### 5.1 EXPERIMENTAL SETUP

We compare TCMRL with FOCAL (Li et al., 2021b), FOCAL++ (Li et al., 2021a), IDAQ (Wang et al., 2023), CSRO (Gao et al., 2023), CORRO (Yuan & Lu, 2022), MACAW (Mitchell et al., 2021) and BOReL (Dorfman et al., 2021) in the Sparse-Point-Robot, Half-Cheetah-Vel, Point-Robot-Wind, Hopper-Rand-Params and Walker-Rand-Params environments, and task sets of the Meta-World ML1 environment (Yu et al., 2019). Notably, for a fair comparison, we employ the same offline datasets for all baselines, leading to some performance biases compared with their original performance. More details about the baselines, the experimental environments and their corresponding datasets are in Appendices E, D and H respectively. The visual analyses are in Appendix G.7, while the analyses about the length of subsequences are in Appendix G.4.

### 5.2 COMPARISON WITH BASELINES

We report experimental results in two forms: one is through table, where we directly compare the results of TCMRL with baselines in a numerical format, as shown in Table 1, and the other is through figure, aiming to showcase the complete processes, as shown in Figure 4. Notably, Table 1 showcases the performance of 20 selected Meta-World ML1 tasks and the complete experimental results are in Appendix G. The comparative results in Figure 4 depict the analysis between TCMRL and baselines in the Sparse-Point-Robot, Half-Cheetah-Vel, Point-Robot-Wind, Hopper-Rand-Params and Walker-Rand-Params environments, and three task sets of Meta-World ML1 (Button-Press-Topdown, Dial-Turn and Reach). Notably, in Table 1, instances denoted by "−" indicate the absence of experimental results for the corresponding baselines within the specified environments. This is due to the lack of support for these experiments and it will not undermine the comparative analysis. Furthermore, all experimental results are averaged across six random seeds and their variances are measured with a 95% bootstrap confidence interval. In summary, TCMRL exhibits superior performance and sample efficiency compared with all baselines in all these environments.

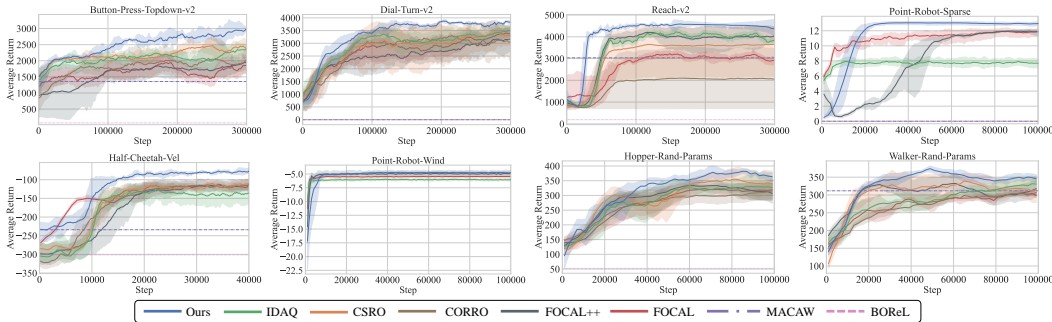

Figure 4: **Comparisons of the effectiveness of adaptation.** The experimental results of TCMRL, IDAQ, CSRO, CORRO, FOCAL++, FOCAL, MACAW and BOReL in the Sparse-Point-Robot, Half-Cheetah-Vel, Point-Robot-Wind, Hopper-Rand-Params and Walker-Rand-Params environments, and three task sets of Meta-World ML1 (Button-Press-Topdown, Dial-Turn and Reach).

In Figure 4, TCMRL demonstrates superior adaptation to unseen target tasks compared with other methods. With our task characteristic extractor to capture the task characteristic information and task contrastive loss to obtain the task contrastive information, TCMRL makes a comprehensive understanding of task information. Then, TCMRL can extract generalizable contexts from trajectories, leading to efficient and effective adaptation to unseen target tasks. In the Button-Press-Topdown, Dial-Turn and Reach task sets, as well as the Hopper-Rand-Params and Walker-Rand-Params environments, TCMRL exhibits faster convergence to superior performance compared with all baselines, despite similar initial performance. Moreover, in the Point-Robot-Wind and Point-Robot-Sparse environments, even when starting with lower initial performance levels, TCMRL outperforms all baselines in terms of convergence speed. In the Half-Cheetah-Vel environment, TCMRL achieves better performance, despite a slightly slower convergence compared with FOCAL. IDAQ, CSRO, CORRO and FOCAL++ exhibit similar sample efficiency to TCMRL but markedly lower performance. Apart from achieving performance similar to FOCAL but significantly lower than TCMRL in the Reach task set, MACAW exhibits poor performance in other environments. Meanwhile, BOReL demonstrates the worst performance in most environments.

## 5.3 ABLATION STUDY

TCMRL employs two main parts: the task characteristic extractor and the task contrastive loss. We build two different variants of the complete framework: one without the task characteristic extractor (w/o TCE) and another without task contrastive loss (w/o TCL). The results in Figure 5 demonstrate that these two variants exhibit similar performance for the three task sets within the Meta-World ML1 (Button-Press-Topdown, Dial-Turn and Reach) and the Half-Cheetah-Vel, Hopper-Rand-Params and Walker-Rand-Params environments, while their sample efficiency and performance are lower than that of TCMRL. Overall, the combined utilization of the task characteristic extractor and the task contrastive loss is essential for capturing comprehensive task information. This enables TCMRL to generate generalizable contexts and achieve efficient and effective adaptation to unseen target tasks.

## 5.4 EFFECTS OF OPTIMIZATION PERSPECTIVES ON THE TASK CHARACTERISTIC EXTRACTOR

To explore the effects of optimization perspectives corresponding to the task characteristic extractor, we build different variants based on TCMRL with only the task characteristic extractor. For simplicity, we abbreviate sparsity in attention weights as sparsity. First, we build variants that make the optimization with one of the three perspectives: with sparsity, with positive reward estimation and with negative reward estimation. Second, we build variants that complete the optimization by excluding one of the three perspectives: without sparsity, without positive reward estimation and without negative reward estimation. We conduct experiments with all these variants in the Half-Cheetah-Vel and Hopper-Rand-Params environments, and the Reach task set within Meta-World ML1. The results in Figure 6(a)-(c) demonstrate that the performance achieved by optimizing only from one of the three perspectives is inferior to that achieved with all perspectives. Moreover, the perspective of positive reward estimation is directly associated with the training of both the task characteristic extractor and the context-based reward estimator, leading to the best performance when employed individually. The perspectives of sparsity in attention weights and negative reward estimation achieve limited performance when applied individually because they only make optimization as constraints. The results in Figure 6(d)-(f) show

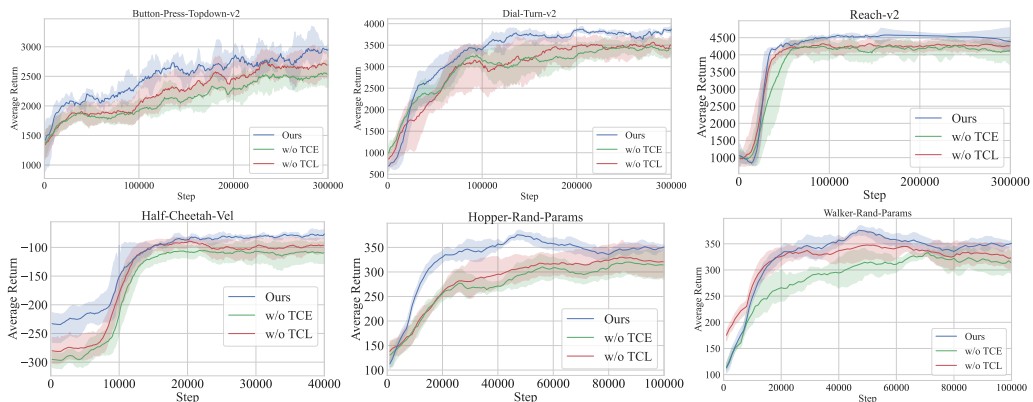

Figure 5: **Ablation experiments on modules.** The variant named w/o TCE removes the task characteristic extractor. The variant named w/o TCL removes the task contrastive loss.

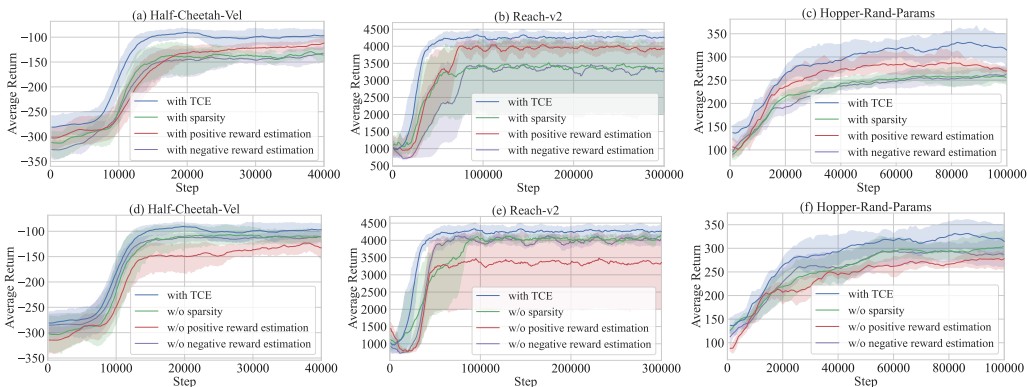

Figure 6: **Effects of the optimization perspectives on the task characteristic extractor.** Variants in (a), (b) and (c) only utilize one of the three perspectives to optimize the task characteristic extractor, while variants in (d), (e) and (f) make optimization with one of the three perspectives removed.

that the removal of any optimization perspective results in a performance decline. Specifically, when the perspectives of sparsity in attention weights or negative reward estimation are removed, the performance is degraded due to missing part of the constraints. When the perspective of positive reward estimation is removed, the performance is limited with two constraints. Overall, the combined utilization of these three perspectives can achieve valid optimization of the task characteristic extractor, while the perspective of positive reward estimation plays a major role and the perspective of negative reward estimation and sparsity in attention weights are effective constraints. More analyses can be found in Appendix G.2.

# 6 CONCLUSION

We propose TCMRL, an offline meta-RL framework that captures comprehensive task information, which includes both task characteristic information and task contrastive information. It leads to contexts with improved generalization, and achieves efficient and effective adaptation to unseen target tasks. Specifically, we propose a task characteristic extractor that identifies and emphasizes transitions, within a trajectory, that are characteristic of a task when generating the context. A context-based reward estimator and a series of specific loss functions are used to optimize the task characteristic extractor and ensure the accurate assignment of attention weights. Moreover, we propose a task contrastive loss to learn task information that distinguishes tasks from one another by considering the overlooked interrelations among transitions from trajectory subsequences. Our experimental evaluations in deterministic continuous control meta-environments demonstrate the superior performance of TCMRL compared with previous offline meta-RL methods.

## REPRODUCIBILITY STATEMENT

Here we detail the efforts that we have made to ensure the reproducibility of our work. Specifically, we provide an anonymous link where the source code of TCMRL is downloadable in Section 5. In Appendix D, we provide detailed descriptions of the environments and task sets used in our work. In Appendix F, we provide detailed descriptions of the method for constructing offline datasets, implementation details, and hyperparameter settings. We also provide the average returns of our offline datasets in Appendix H.

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

# A   PSEUDO-CODE OF TCMRL

We present our meta-training process in Algorithm 1 and our meta-testing process in Algorithm **??**.

---

**Algorithm 1** TCMRL meta-training.

---

**Input:** The set of offline datasets $\mathbb{D} = \{\mathcal{D}_i\}_{i=1}^{n_{task}}$; Context encoder $e(h_i^t)$; Task characteristic extractor $q(c_i^t, \bar{c}_i)$; Context-based reward estimator $\hat{r}(s_i^t, a_i^t, c_i)$; Task contrastive loss $L_{TCL}$; Context-based policy $\pi(a_i^t | s_i^t, c_i)$; Q-function $Q$.

1: **while** not done **do**
2:   **for** step in training steps **do**
3:     Sample $D_i \sim \mathbb{D}$ corresponding to $\mathcal{T}_i$ and sample historical trajectory $\boldsymbol{h}_i$ from it
4:     Extract $\{c_i^t\}_{t=1}^T$ from $\{h_i^t\}_{t=1}^T$ through $e(h_i^t)$ (Eq. 1)
5:     Compute $L_{TCL}$ (Eq. 11)
6:     Compute $\bar{c}_i$ (Eq. 2)
7:     Compute $(score_i^t)_{t=1}^T$ for $\{c_i^t\}_{t=1}^T$ with $\bar{c}_i$ and $q(c_i^t, \bar{c}_i)$ (Eq. 3)
8:     Compute $(w_i^t)_{t=1}^T$ for $\{c_i^t\}_{t=1}^T$ through softmax function and $(score_i^t)_{t=1}^T$ (Eq. 4)
9:     Compute $c_i$ with $\{c_i^t\}_{t=1}^T$ and $(w_i^t)_{t=1}^T$ (Eq. 5)
10:     Update $e(h_i^t)$, $q(c_i^t, \bar{c}_i)$ and $\hat{r}(s_i^t, a_i^t, c_i)$ to minimize $L_{TCE}^{spa}$, $L_{TCE}^{pos}$ and $L_{TCE}^{neg}$ (Eq. 6, Eq. 7 and Eq. 8)
11:     Update $\pi(a_i^t | s_i^t, c_i)$ and $Q$ with offline RL algorithm SAC (Haarnoja et al., 2018)
12:   **end for**
13: **end while**

---

**Algorithm 2** TCMRL meta-testing.

---

**Input:** The set of unseen target tasks $\mathbb{T}^*$; Context encoder $e(h_i^t)$; Task characteristic extractor $q(c_i^t, \bar{c}_i)$; Learned context-based policy $\pi(a_i^t | s_i^t, c_i)$; Random explore step $t_r$.

1: **for** each unseen target task $\mathcal{T}_j \sim \mathbb{T}^*$ **do**
2:   $\boldsymbol{h}_j = \{\}$
3:   **for** $t = 0, ..., T-1$ **do**
4:     **if** $t < t_r$ **then**
5:       Agent randomly samples an action $a_j^t$ to collect transition $h_j^t = (s_j^t, a_j^t, r_j^t, s'^t_j)$
6:     **else**
7:       Compute context $c_j$ with $e(h_i^t)$ and $q(c_i^t, \bar{c}_i)$ (Eq. 1, Eq. 2, Eq. 3, Eq. 4 and Eq. 5)
8:       Agent uses $\pi(a_i^t | s_i^t, c_i)$ to roll out $h_j^t = (s_j^t, a_j^t, r_j^t, s'^t_j)$
9:     **end if**
10:     $\boldsymbol{h}_j = \boldsymbol{h}_j \cup h_j^t$
11:   **end for**
12:   Compute context $c_j$ with $e(h_i^t)$ and $q(c_i^t, \bar{c}_i)$ (Eq. 1, Eq. 2, Eq. 3, Eq. 4 and Eq. 5)
13:   Roll out $\pi(a_i^t | s_i^t, c_i)$ for evaluation
14: **end for**

---

# B   PRELIMINARIES OF META-LEARNING

We choose the standard supervised meta-learning to illustrate the concept of meta-learning (see, e.g., (Finn et al., 2017)). We assume tasks $\mathcal{T}_i$ are sampled from a distribution of tasks $p(\mathcal{T})$. The problem setting of the meta-learning consists of two phases: the meta-training phase and the meta-testing phase. These two phases confront distinct sets of tasks, with no overlap between the tasks they encounter. During the meta-training phase, a meta-model is learned through a set of meta-training tasks $\mathbb{T}$. We sample a set of meta-training data $\mathbb{D}$ from these tasks. For a particular task $\mathcal{T}_i$, the corresponding meta-training data $\mathcal{D}_i$ consists of a subset for training $(x_i, y_i)$ and a subset for testing, while $x_i = (x_i^1, x_i^2, ..., x_i^T)$ and $y_i = (y_i^1, y_i^2, ..., y_i^T)$ are sampled from $p(x_i, y_i | \mathcal{T}_i)$, and $x_i^* = (x^{*1}_i, x^{*2}_i, ..., x^{*T}_i)$ and $y^*_i = (y^{*1}_i, y^{*2}_i, ..., y^{*T}_i)$ are sampled from $p(x_i^*, y^*_i | \mathcal{T}_i)$. During the meta-testing phase, the learned meta-model is utilized to address a set of unseen target tasks $\mathbb{T}^*$ and tries to achieve efficient and effective adaptation. We denote the meta-parameters learned during the meta-training phase as $\theta$ and the task-specific parameters computed based on the meta-training tasks as $\phi$.

Following Grant et al. (2018) and Gordon et al. (2019), we assess meta-learning algorithms that aim to use the meta-training data $\mathbb{D}$ corresponding to the set of meta-training tasks $\mathbb{T}$ to maximize conditional likelihood $q(\hat{y}^* = y^* | x^*, \theta, \mathbb{D})$, which is related to three distributions: $q(\theta | \mathbb{D})$ that generates the distribution of the meta-parameters $\theta$ from the meta-training data $\mathbb{D}$, $q(\phi | \mathcal{D}_i, \theta)$ that generate the distribution of the task-specific parameters $\phi$ and $q(\hat{y}^* | x^*, \phi, \theta)$ that is the predictive distribution. The learning objective of these distributions is as follows:

$$-\frac{1}{N} \sum_i \mathbb{E}_{q(\theta|\mathbb{D})q(\phi|\mathcal{D}_i,\theta)} \left[ \frac{1}{T} \sum_{(x^*,y^*) \in \mathcal{D}_i} \log q(\hat{y}^* = y^* | x^*, \phi, \theta) \right]. \tag{13}$$

Meta-learning algorithms can be primarily categorized into two kinds of distinct algorithms: optimization-based algorithms and context-based algorithms. Specifically, MAML (Finn et al., 2017) is a classic optimization-based meta-learning algorithm. Within MAML, $\theta$ and $\phi$ denote the weights of the predictor network, $q(\phi | \mathcal{D}_i, \theta)$ is a delta function that is positioned at a location determined through gradient optimization, and $\phi$ parameterizes the predictor network $q(\hat{y}^* | x^*, \phi)$. Moreover, it utilizes the meta-training data $\mathcal{D}_i$ and the parameter $\theta$ in the predictor model for determining the task-specific parameter $\phi$, and this process is as follows:

$$\phi = \theta + \frac{\alpha}{T} \sum_{(x,y) \in \mathcal{D}_i} \nabla_\theta \log q(y|x, \phi = \theta). \tag{14}$$

Meanwhile, the conditional neural processes (CNP) (Garnelo et al., 2018) is a notable context-based algorithm, which defines $q(\phi | \mathbb{D}, \theta)$ as a mapping from $\mathbb{D}$ to the parameter $\phi$. Features $e(\mathbb{D})$ extracted from the meta-training data are aggregated through a network $agg_\theta(\cdot)$, and the output is computed through $\phi = agg_\theta \cdot e(\mathbb{D})$. Subsequently, the parameter $\theta$ defines a predictor network that inputs $\phi$ and $x^*$ and outputs the prediction of the distribution $q(\hat{y}^* | x^*, \phi, \theta)$.

## C  PRELIMINARIES OF CONTEXT-BASED OFFLINE META-RL

We assume that context-based offline meta-RL corresponds to a set of tasks consisting of a series of meta-training tasks and a series of meta-testing tasks (unseen target tasks). These tasks within this set shares the same state space $\mathcal{S}$ and action space $\mathcal{A}$, but exhibit variations in their transition dynamics $p(s_i^{t+1} | s_i^t, a_i^t)$ or reward functions $r(s_i^t, a_i^t)$. Moreover, a distribution of these tasks is modeled as joint distribution of transition dynamics $p(s_i^{t+1} | s_i^t, a_i^t)$ and reward functions $r(s_i^t, a_i^t)$, with the following form:

$$p(\mathcal{T}) := p(p(s_i^{t+1} | s_i^t, a_i^t), r(s_i^t, a_i^t)) = p(p(s_i^{t+1} | s_i^t, a_i^t)) p(r(s_i^t, a_i^t)). \tag{15}$$

This task distribution corresponds to a series of MDPs, and a meta-policy designed by context-based offline meta-RL methods aims to perform well across all these MDPs. These MDPs are formed as POMDPs since they consider the task information of each task to be the unobservable part. Consequently, a context encoder $e(\cdot)$ is utilized to map the task information of the historical trajectory $\boldsymbol{h}$ that corresponds to the task $\mathcal{T}$ to a representation of the context $c \in C$, where $C$ is the space of contexts. The form of the augmented state is as follows:

$$\mathcal{S}_{\text{aug}} \leftarrow \mathcal{S} \times \mathcal{C}, \quad s_{\text{aug}} \leftarrow \text{concat}(s, c). \tag{16}$$

This set of MDPs is also defined as task-augmented MDP (TA-MDP) (Li et al., 2021b;a).

Previous context-based offline meta-RL methods (Li et al., 2021b; Rakelly et al., 2019; Wang et al., 2023) typically obtain task information of task $\mathcal{T}_i$ by aggregating transitions from the historical trajectory $\boldsymbol{h}_i^{1:t} = \{s_i^1, a_i^1, r_i^1, s_i^2 ..., s_i^t, a_i^t, r_i^t, s_i^{t+1}\}$ into a representation of the continuous latent space of contexts $\mathcal{C}$. These methods have proved that the quality of contexts, or the ability of the context encoder to extract task information from historical trajectories, directly influences the performance of the meta-policy and its adaptation to unseen target tasks. In addition, as a traditional and successful context-based offline meta-RL method, probabilistic representations for actor-critic RL (PEARL) (Rakelly et al., 2019) generates contexts $c_i$ in the form of vectors. Moreover, the complete process of adaptation to unseen target tasks involves sampling the vector $c_i$ from the corresponding probabilistic distribution $q_e(c_i | \boldsymbol{h}_i)$, which is parameterized by an encoder $e$. Here, $\boldsymbol{h}_i$ is a complete historical trajectory corresponding to the episode of task $\mathcal{T}_i$. Specifically, the context encoder is implemented by a neural network and the input historical trajectory consists of a series of transitions $h_i^t = (s_i^t, a_i^t, r_i^t, s_i^{t+1})$. Additionally, the context $c_i$ is one of the inputs of the context-based policy $\pi(a_i^t | s_i^t, c_i)$ for making action decisions.

## D EXPERIMENTAL ENVIRONMENTS

- **Sparse-Point-Robot**. The Sparse-Point-Robot environment consists of a 2D navigation problem, simulated by the MuJoCo physics simulator and introduced in PEARL (Rakelly et al., 2019). In this environment setting, each task involves guiding the agent from the origin to a specific goal position situated on the unit circle centered at the origin. The non-sparse reward is defined as the negative of the distance between the current location and the goal position of the agent. In the case of a sparse-reward scenario, the reward is set to 0 when the agent is outside a neighborhood surrounding the goal, which is controlled by the goal radius. Conversely, when the agent is inside this neighborhood, it receives a reward of 1 minus the distance at each step, yielding a positive value. We use the sparse-reward scenario.
- **Half-Cheetah-Vel**. The Half-Cheetah-Vel environment serves as a multi-task MuJoCo benchmark wherein tasks exhibit variations in their reward functions. Specifically, definitions of these tasks are revolved around the specification of the target velocity of the agent. The distribution of the target velocity follows a uniform distribution denoted as $U[0, v_{max}]$.
- **Point-Robot-Wind**. The Point-Robot-Wind environment is a variant of the 2D navigation environment called Point-Robot. In this variant, each task solely differs in their transition functions, while sharing the same reward function. Specifically, each task is characterized by a distinct wind, which is uniformly sampled from $[-l, l]^2$. Consequently, whenever the agent takes a step, it undergoes a drift determined by the corresponding wind.
- **Hopper-Rand-Params**. The Hopper-Rand-Params environment controls the forward movement of a single-legged robot. Tasks encompass diverse aspects such as body mass, body inertia, joint damping, and friction. Each parameter is determined by the default value multiplied by a coefficient randomly selected from the range $[1.5^{-3}, 1.5^3]$. The state space is $\mathbb{R}^{11}$ and the action space is $[-1, 1]^3$. Meanwhile, the reward function comprises forward velocity and bonuses for staying alive and controlling costs.
- **Walker-Rand-Params**. The Walker-Rand-Params environment controls the forward movement of a bipedal robot. Similar to the Hopper-Rand-Params environment, each parameter is determined using the same method. Meanwhile, the reward function mirrors that of Hopper-Rand-Params. The state space encompasses $\mathbb{R}^{17}$, while the action space is $[-1, 1]^6$.
- **Meta-World ML1** (Yu et al., 2019). The Meta-World ML1 environment comprises 50 robot arm manipulation task sets. Specifically, each task entails controlling a robotic arm to accomplish a given objective, as evident from their descriptive names such as Button-Press-Topdown, Dial-Turn, Reach, and Window-Open. These tasks closely resemble real-world scenarios and actions.

Additionally, in the meta-RL environments we employed, each task is characterized by distinct goals. In the Sparse-Point-Robot and Half-Cheetah-Vel environments, their task sets both consist of 100 tasks, of which 80 tasks are designated as meta-training tasks and 20 tasks are designated as meta-testing tasks. In the Point-Robot-Wind and Meta-World ML1 environments, their task sets both comprise 50 tasks, wherein 40 tasks are meta-training tasks and 10 tasks are meta-testing tasks. In the Hopper-Rand-Params and Walker-Rand-Params environments, their task sets both consist of 40 tasks, while 30 tasks are meta-training tasks and 10 tasks are meta-testing tasks. Notably, all these MuJoCo environments and Meta-World ML1 task sets have MIT licenses.

## E BASELINES

- **FOCAL** (Li et al., 2021b). FOCAL introduces behavior regularization to the learned policy framework while utilizing a deterministic context encoder for efficient task inference. Furthermore, it incorporates a novel negative-power distance metric within a bounded context embedding space, enabling gradient propagation that is decoupled from the Bellman backup process. Specifically, it treats all online experiences as effective data for generating contexts.
- **FOCAL++** (Li et al., 2021a). FOCAL++ is a framework that has been built upon and is expanding the foundation of FOCAL. It aims to address the problem of MDP ambiguity (Li et al., 2020), which is due to the biased distribution of the fixed datasets, through attention mechanism and contrastive learning objectives.
- **IDAQ** (Wang et al., 2023). IDAQ is a framework that extends the foundations of FOCAL. It leverages a return-based uncertainty quantification to generate context within the in-distribution. Additionally, it utilizes effective task belief inference methods to tackle new tasks.
- **CSRO** (Gao et al., 2023). CSRO is an approach that addresses the context shift problem with only offline datasets by minimizing the influence of policy in context during both the

meta-training and meta-test phases. Specifically, a max-min mutual information representation learning mechanism is designed to diminish the impact of the behavior policy on task representations during the meta-training phase. The non-prior context collection strategy is introduced to reduce the effect of the exploration policy during the meta-testing phase.

- **CORRO** (Yuan & Lu, 2022). CORRO is a context-based meta-RL framework for addressing the change of behavior policies. It aims to learn how to obtain robust task representations through contrastive learning.
- **MACAW** (Mitchell et al., 2021). MACAW is an optimization-based meta-learning algorithm that adheres to the offline meta-RL setting. In addition, it employs the simple and supervised regression objectives for both the inner and outer loops of meta-training, ensuring effective performance.
- **BOReL** (Dorfman et al., 2021). BOReL is an algorithm that addresses the challenges of the offline meta-RL from the view of Bayesian RL (BRL). Its main objective is to learn a Bayes-optimal policy using offline data. Moreover, it extends the VariBAD BRL approach (Zintgraf et al., 2020) by incorporating an off-policy learning framework and an adaptive neural belief estimate and focuses on planning an exploration strategy that maximizes information gain based on the learned belief model.

Notably, all these baselines have MIT licenses.

## F  IMPLEMENTATION DETAILS

### F.1  OFFLINE DATA COLLECTIONS

To ensure a fair comparison, we follow the same approach as IDAQ in generating the offline datasets, which are used during the meta-training phase (see Appendix H).

For each training task, we employ SAC (Haarnoja et al., 2018) to train an agent and store the policy at various training times as the behavior policy. Each policy is employed to roll out 50 trajectories in the corresponding environment to construct offline datasets. Notably, this is a common approach for constructing offline datasets in the field of offline meta-RL (Li et al., 2021b;a; Yuan & Lu, 2022; Wang et al., 2023; Gao et al., 2023).

### F.2  EXPERIMENTAL DETAILS

Our experiments are conducted on a machine with NVIDIA GeForce RTX 2080 Ti and implemented with PyTorch.

TCMRL employs the Adam optimizer (Kingma & Ba, 2015) with a learning rate of $3e-4$ for the policy, Q-network, V-network and context encoder, and a learning rate of $1e-4$ for the dual critic. We set the batch size to 256, and the discount factor to 0.99. We implement our task characteristic extractor with a multi-layer perceptron (MLP) neural network architecture. Each hidden layer is a fully connected layer with 256 units and the activation function is the sigmoid function. The context-based reward estimator is also implemented with an MLP architecture, while each hidden layer is a fully connected layer with 256 units.

As depicted in Figure 4, we train 100000 steps in the Point-Robot-Wind and Point-Robot-Sparse environments, and 40000 steps in the Half-Cheetah-Vel environment. Moreover, for most of Meta-World ML1 tasks, such as "Button-Press-Topdown", "Dial-Turn" and "Reach", we train them for 300000 steps. However, it has been observed that training with excessively long steps leads to performance degradation for some tasks, such as "Door-Close". Therefore, based on the observations, we reduce the number of training steps for them. Moreover, because the hyperparameter $K$ used in discovering interrelations among transitions is crucial and sensitive, we carefully set it for each task to ensure optimal performance. Specifically, we set $K$ to 5 for Point-Robot-Wind, 2 for Point-Robot-Sparse and 4 for Half-Cheetah-Vel. Additionally, on the Meta-World ML1 task set, taking a few tasks as examples, we set $K$ to 6 for "Reach", "Basketball" and "Bin-Picking", 4 for "Button-Press-Topdown" and "Dial-Turn" and 8 for "Box-Close". Furthermore, we set the dimension of the context to 20 in most environments, while it is set to 40 in the Walker-Rand-Params environment.

F.3  DETAILS OF THE CONTEXT-BASED REWARD ESTIMATOR

Actually, we employ two different processing levels to handle environments with sparse and dense rewards respectively. When handling reward-dense environments, the context-based reward estimator $\hat{r}(s_i^t, a_i^t, c_i)$ operates at the level of transitions within a trajectory as Eq. 7 and Eq. 8, since there is rich reward information at each trajectory step.

While meeting reward-sparse environments, the operation of the context-based reward estimator shifts to the level of trajectories, since there is limited reward information in only a few trajectory steps. In such cases, we input state-action pairs of the entire trajectory, and the context to estimate the cumulative reward of the entire trajectory. The learning objective of positive reward estimation $L_{TCE}^{pos}$ in reward-sparse environments is as follows:

$$L_{TCE}^{pos} = (\hat{r}(s'^1_i, a'^1_i, ..., s'^T_i, a'^T_i, c_i) - \sum_{t=1}^{T} r'^t_i)^2. \tag{17}$$

Meanwhile, the learning objective of negative reward estimation $L_{TCE}^{neg}$ in reward-sparse environments is as follows:

$$L_{TCE}^{neg} = (\hat{r}_{reverse}(s'^1_i, a'^1_i, ..., s'^T_i, a'^T_i, c_i^{neg}) - \sum_{t=1}^{T} r'^{t\,neg}_i)^2. \tag{18}$$

We conduct experiments in the Sparse-Point-Robot environment, which is reward-sparse and these results can be found in Table 1 and Figure 4.

# G  MORE EXPERIMENTAL RESULTS

## G.1  COMPLETE EXPERIMENTAL RESULTS

Table 2 shows the experimental results in 50 Meta-World ML1 task sets and MuJoCo tasks. Additionally, it is worth mentioning that all Meta-World ML1 tasks are originally named with a "-v2" suffix. However, for the sake of conciseness, we have omitted this suffix in our presentation. Overall, TCMRL demonstrates superior performance compared with all baselines, achieving more efficient and effective adaptation to unseen target tasks. Notably, FOCAL++ utilizes attention mechanisms at both the sequence-wise and batch-wise. We conduct comparisons not only with the complete FOCAL++ but also separately with these two different attention mechanisms. Meanwhile, CORRO employs two distinct methods for the generation of negative samples: one leverages the condition variational auto-encoder (CVAE), while the other utilizes the reward randomization (RR). The results of CORRO presented in Table 1 and Table 2 represent the maximum performance attained across both CORRO with CVAE and CORRO with RR, serving as a comprehensive result for comparison. The comparative results between TCMRL and FOCAL++ can be found in Appendix G.5 while results between TCMRL and CORRO can be found in Appendix G.6.

## G.2  ADDITIONAL ANALYSIS OF
### EFFECTS OF OPTIMIZATION PERSPECTIVES ON THE TASK CHARACTERISTIC EXTRACTOR

To further analyze the effects of different optimization perspectives on the task characteristic extractor (sparsity in attention weights, positive reward estimation and negative reward estimation), we conduct experiments in more environments to explore their individual effects. The results in Figure 7 and Figure 8 align with the conclusions drawn in Section 5.4 in the Walker-Rand-Params environment, and the Button-Press-Topdown and Dial-Turn task sets within Meta-World ML1. Additionally, some special cases required further analysis. As mentioned in Section 5.4, the perspective of positive reward estimation primarily involves direct training of the context-based reward estimator and the task characteristic extractor, demonstrating significant importance. Meanwhile, the perspectives of both sparsity in attention weights and negative reward estimation mainly serve as constraints. The performance and sample efficiency shown in Figure 7 present the results when optimization is performed solely from one of three perspectives. The variant with positive reward estimation shows the best performance in these environments among all variants that only optimize the task characteristic extractor from a single perspective because of its effect on training. The variant with sparsity exhibits significant performance fluctuations in the Button-Press-Topdown and Dial-Turn task sets within Meta-World ML1, since the limited effectiveness of optimization solely from the single perspective of constraint. The variant with negative reward estimation demonstrates relatively stable performance in the Button-Press-Topdown task set, whereas it still exhibits significant performance fluctuations in the Dial-Turn task set. This is due to

Table 2: Comparison between IDAQ, CSRO, CORRO, FOCAL++, FOCAL, MACAW, and BOReL with online adaptation and TCMRL.

| Task set/Environment | TCMRL (ours) | IDAQ | CSRO | CORRO | FOCAL++ | FOCAL | MACAW | BOReL |
|---|---|---|---|---|---|---|---|---|
| Assembly | **0.56±0.15** | 0.55±0.13 | 0.26±0.18 | 0.38±0.08 | 0.51±0.11 | 0.28±0.05 | 0.33±0.01 | 0.04±0.00 |
| Basketball | **0.82±0.11** | 0.64±0.15 | 0.58±0.10 | 0.57±0.04 | 0.71±0.25 | 0.41±0.24 | 0.00±0.00 | 0.00±0.00 |
| Bin-Picking | **0.65±0.10** | 0.53±0.16 | 0.57±0.13 | 0.47±0.14 | 0.51±0.24 | 0.31±0.21 | 0.66±0.11 | 0.00±0.00 |
| Box-Close | **0.62±0.09** | 0.51±0.11 | 0.51±0.02 | 0.60±0.03 | 0.44±0.03 | 0.15±0.09 | 0.36±0.11 | 0.05±0.01 |
| Button-Press-Topdown | **0.81±0.12** | 0.57±0.11 | 0.66±0.09 | 0.55±0.14 | 0.51±0.10 | 0.45±0.10 | 0.38±0.36 | 0.02±0.02 |
| Button-Press-Topdown-Wall | **0.47±0.02** | 0.43±0.03 | 0.37±0.02 | 0.35±0.05 | 0.42±0.02 | 0.40±0.07 | 0.05±0.02 | 0.05±0.01 |
| Button-Press | **0.81±0.05** | 0.74±0.08 | 0.69±0.05 | 0.72±0.04 | 0.79±0.05 | 0.68±0.14 | 0.02±0.01 | 0.01±0.01 |
| Button-Press-Wall | **1.07±0.03** | 1.04±0.04 | 1.04±0.01 | 1.02±0.04 | 0.98±0.07 | 0.99±0.06 | 0.02±0.00 | 0.01±0.00 |
| Coffee-Button | **0.83±0.12** | 0.73±0.14 | 0.79±0.08 | 0.77±0.04 | 0.75±0.16 | 0.66±0.16 | 0.15±0.13 | 0.02±0.00 |
| Coffee-Pull | **0.51±0.05** | 0.40±0.05 | 0.48±0.01 | 0.43±0.04 | 0.32±0.04 | 0.23±0.04 | 0.19±0.12 | 0.00±0.00 |
| Coffee-Push | **1.27±0.08** | 1.22±0.13 | 1.22±0.01 | 1.17±0.16 | 1.00±0.05 | 0.64±0.07 | 0.01±0.01 | 0.00±0.00 |
| Dial-Turn | **0.98±0.01** | 0.91±0.05 | 0.81±0.09 | 0.87±0.07 | 0.80±0.13 | 0.84±0.09 | 0.00±0.00 | 0.00±0.00 |
| Disassemble | **0.59±0.13** | 0.41±0.14 | 0.56±0.06 | 0.49±0.06 | 0.32±0.08 | 0.25±0.04 | 0.05±0.00 | 0.04±0.00 |
| Door-Close | **1.01±0.00** | 0.99±0.00 | 0.74±0.18 | 0.98±0.01 | 1.01±0.00 | 0.97±0.01 | 0.00±0.00 | 0.37±0.19 |
| Door-Lock | **0.99±0.00** | 0.97±0.01 | 0.94±0.02 | 0.89±0.05 | 0.96±0.00 | 0.90±0.02 | 0.25±0.11 | 0.14±0.00 |
| Door-Unlock | **1.18±0.02** | 1.11±0.02 | 1.13±0.01 | 1.15±0.01 | 1.11±0.02 | 0.97±0.03 | 0.11±0.01 | 0.13±0.03 |
| Door-Open | **1.00±0.00** | 0.94±0.02 | 0.98±0.00 | 0.91±0.05 | 0.92±0.01 | 0.76±0.13 | 0.06±0.01 | 0.11±0.01 |
| Drawer-Close | **1.01±0.01** | 0.99±0.02 | 1.00±0.01 | 0.94±0.02 | 0.97±0.01 | 0.96±0.04 | 0.53±0.50 | 0.00±0.00 |
| Drawer-Open | **0.90±0.03** | 0.82±0.06 | 0.54±0.12 | 0.74±0.04 | 0.84±0.05 | 0.64±0.10 | 0.11±0.02 | 0.10±0.00 |
| Faucet-Close | **1.13±0.01** | 1.12±0.01 | 1.10±0.01 | 1.11±0.01 | 1.11±0.00 | 1.06±0.02 | 0.07±0.01 | 0.13±0.03 |
| Faucet-Open | **1.08±0.02** | 1.05±0.02 | 1.05±0.01 | 1.07±0.00 | 1.06±0.00 | 1.01±0.02 | 0.08±0.04 | 0.12±0.05 |
| Hammer | **0.85±0.06** | 0.83±0.06 | 0.77±0.07 | 0.79±0.05 | 0.83±0.04 | 0.58±0.07 | 0.10±0.01 | 0.09±0.01 |
| Hand-Insert | **0.72±0.05** | 0.63±0.04 | 0.64±0.02 | 0.63±0.13 | 0.56±0.06 | 0.29±0.07 | 0.02±0.01 | 0.00±0.00 |
| Handle-Press-Side | **0.96±0.02** | 0.88±0.02 | 0.57±0.03 | 0.33±0.10 | 0.83±0.04 | 0.77±0.10 | 0.49±0.41 | 0.02±0.01 |
| Handle-Press | 0.77±0.06 | 0.88±0.05 | 0.34±0.03 | 0.31±0.07 | 0.90±0.02 | 0.87±0.02 | 0.28±0.10 | 0.01±0.00 |
| Handle-Pull-Side | **0.31±0.08** | 0.12±0.04 | 0.14±0.02 | 0.10±0.06 | 0.26±0.07 | 0.11±0.08 | 0.00±0.00 | 0.00±0.00 |
| Handle-Pull | **0.92±0.02** | 0.89±0.02 | 0.35±0.27 | 0.63±0.19 | 0.76±0.04 | 0.66±0.03 | 0.00±0.00 | 0.00±0.00 |
| Lever-Pull | **0.86±0.02** | 0.84±0.02 | 0.79±0.03 | 0.81±0.03 | 0.62±0.06 | 0.72±0.07 | 0.20±0.16 | 0.05±0.00 |
| Peg-Insert-Side | **0.45±0.05** | 0.30±0.04 | 0.27±0.14 | 0.36±0.10 | 0.19±0.07 | 0.08±0.03 | 0.00±0.00 | 0.00±0.00 |
| Peg-Unplug-Side | **0.69±0.01** | 0.56±0.07 | 0.22±0.04 | 0.22±0.07 | 0.50±0.03 | 0.19±0.09 | 0.00±0.00 | 0.00±0.00 |
| Pick-Out-Of-Hole | **0.71±0.06** | 0.25±0.25 | 0.54±0.13 | 0.52±0.14 | 0.29±0.17 | 0.15±0.16 | 0.59±0.06 | 0.00±0.00 |
| Pick-Place | **0.32±0.09** | 0.19±0.03 | 0.11±0.03 | 0.25±0.05 | 0.14±0.03 | 0.07±0.02 | 0.05±0.05 | 0.00±0.00 |
| Pick-Place-Wall | **0.43±0.15** | 0.28±0.12 | 0.36±0.18 | 0.34±0.24 | 0.18±0.06 | 0.09±0.04 | 0.39±0.25 | 0.00±0.00 |
| Plate-Slide-Back-Side | **0.97±0.03** | 0.97±0.02 | 0.31±0.16 | 0.64±0.22 | 0.96±0.02 | 0.77±0.20 | 0.02±0.01 | 0.01±0.00 |
| Plate-Slide-Back | **0.98±0.02** | 0.96±0.02 | 0.29±0.13 | 0.80±0.04 | 0.89±0.04 | 0.58±0.06 | 0.21±0.17 | 0.01±0.00 |
| Plate-Slide-Side | 1.07±0.08 | 1.07±0.08 | 0.98±0.03 | 0.99±0.01 | 0.99±0.07 | 0.70±0.14 | 0.00±0.00 | 0.00±0.00 |
| Plate-Slide | **1.01±0.02** | 1.01±0.03 | 1.00±0.00 | 0.91±0.02 | 0.92±0.01 | 0.83±0.09 | 0.01±0.00 | 0.01±0.00 |
| Push-Back | **0.58±0.04** | 0.52±0.05 | 0.17±0.10 | 0.21±0.07 | 0.21±0.05 | 0.16±0.04 | 0.00±0.00 | 0.00±0.00 |
| Push | **0.64±0.12** | 0.55±0.10 | 0.60±0.07 | 0.57±0.08 | 0.62±0.09 | 0.34±0.14 | 0.28±0.19 | 0.00±0.00 |
| Push-Wall | **0.77±0.11** | 0.71±0.15 | 0.71±0.02 | 0.69±0.07 | 0.74±0.07 | 0.43±0.06 | 0.23±0.18 | 0.00±0.00 |
| Reach | **0.92±0.03** | 0.85±0.03 | 0.75±0.20 | 0.43±0.36 | 0.87±0.04 | 0.62±0.05 | 0.63±0.04 | 0.04±0.01 |
| Reach-Wall | **0.94±0.05** | 0.93±0.05 | 0.91±0.01 | 0.84±0.07 | 0.92±0.03 | 0.53±0.18 | 0.82±0.02 | 0.06±0.00 |
| Shelf-Place | **0.84±0.12** | 0.70±0.18 | 0.54±0.05 | 0.59±0.15 | 0.53±0.04 | 0.32±0.11 | 0.01±0.01 | 0.00±0.00 |
| Soccer | **0.60±0.06** | 0.44±0.04 | 0.54±0.11 | 0.58±0.07 | 0.29±0.03 | 0.11±0.03 | 0.38±0.31 | 0.04±0.02 |
| Stick-Pull | **0.37±0.08** | 0.32±0.06 | 0.16±0.01 | 0.33±0.04 | 0.33±0.04 | 0.17±0.07 | 0.00±0.00 | 0.00±0.00 |
| Stick-Push | **0.85±0.05** | 0.73±0.09 | 0.71±0.06 | 0.76±0.05 | 0.83±0.05 | 0.46±0.15 | 0.17±0.17 | 0.00±0.0 |
| Sweep-Into | **0.66±0.02** | 0.61±0.06 | 0.53±0.02 | 0.44±0.03 | 0.58±0.03 | 0.33±0.05 | 0.00±0.00 | 0.01±0.00 |
| Sweep | **0.84±0.03** | 0.77±0.04 | 0.75±0.06 | 0.53±0.18 | 0.37±0.11 | 0.32±0.08 | 0.20±0.20 | 0.00±0.00 |
| Window-Close | **0.95±0.01** | 0.93±0.01 | 0.93±0.02 | 0.92±0.01 | 0.94±0.01 | 0.79±0.01 | 0.54±0.44 | 0.03±0.00 |
| Window-Open | **0.98±0.02** | 0.96±0.02 | 0.50±0.03 | 0.48±0.02 | 0.88±0.03 | 0.81±0.07 | 0.15±0.11 | 0.03±0.00 |
| Sparse-Point-Robot | **12.98±0.29** | 7.74±0.68 | – | – | 11.59±0.15 | 11.66±0.46 | 0.00±0.00 | 0.00±0.00 |
| Half-Cheetanh-Vel | **-79.7±11.3** | -133.4±23.9 | -114.5±14.0 | -113.2±17.2 | -116.7±14.9 | -117.7±13.6 | -234.0±23.5 | -301.4±36.8 |
| Point-Robot-Wind | **-4.75±0.26** | -6.03±0.22 | – | – | -4.89±0.31 | -5.46±0.26 | – | – |
| Hopper-Rand-Params | **368.62±10.37** | 325.74±27.09 | 331.65±33.62 | 293.32±17.49 | 318.86±20.14 | 314.41±29.00 | – | 52.82 |
| Walker-Rand-Params | **354.97±19.72** | 324.04±31.40 | 316.81±16.31 | 301.49±5.06 | 313.02±24.22 | 303.07±4.28 | 311.68 | 269.74 |

the diversity among task sets. The performance and sample efficiency shown in Figure 8 illustrates the results when optimization under one of three perspectives is removed. Compared with optimizing the task characteristic extractor solely from one of three perspectives, the variants without sparsity and without negative reward estimation exhibit superior performance. The variant without positive reward estimation implies optimization based on the perspectives of sparsity in attention weights and negative reward estimation, which serve as constraints. It demonstrates relatively stable performance in the Button-Press-Topdown task set, but significant performance fluctuations persist in the Dial-Turn task set. This phenomenon aligns with our previous analysis. Due to variations across different task sets, optimization with both two constraints may stabilize the task characteristic extractor to assign attention weights to transitions, within a trajectory, that are characteristic of a task. However, significant fluctuations may persist in some environments or task sets. In general, the combined effect of these three perspectives is indispensable and allows the effective optimization of the task characteristic extractor.

### G.3 ANALYSIS OF NEGATIVE REWARD

To generate generalizable contexts with task characteristic information, we train the task characteristic extractor (TCE) from three different perspectives. One of these perspectives is the negative reward estimation, where we introduce negative reward $r'^{t^{neg}}_i$ by adding random noise to the reward $r'^t_i$ ($r'^{t^{neg}}_i = r'^t_i + r^{noise}$). In addition, we conduct experiments where we directly select the maximum reward $r'^{max}_i$ or the minimum reward $r'^{min}_i$ from the trajectory $\boldsymbol{h}'_i$ and use it as the negative reward with larger biases for all transitions $h'^t_i$ within the trajectory $\boldsymbol{h}'_i$. Based on the three methods, we can construct corresponding variants TCE with random rewards, TCE with the maximum reward and

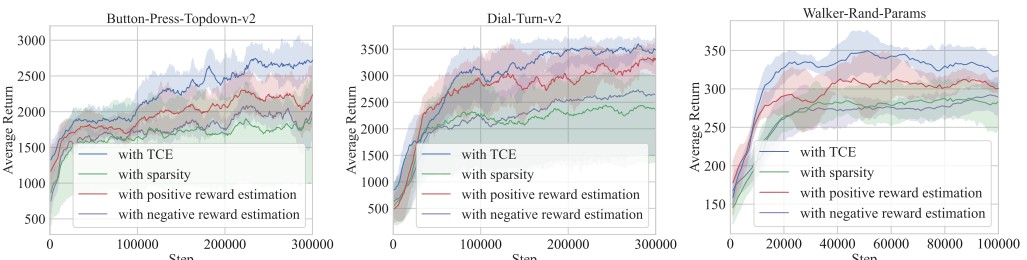

Figure 7: **Additional analysis of the effects of single optimization perspectives on the task characteristic extractor.** Experiments in two task sets of Meta-World ML1 (Button-Press-Topdown and Dial-Turn), and the Walker-Rand-Params environment for exploring the separate effect on optimization perspectives on the task characteristic extractor.

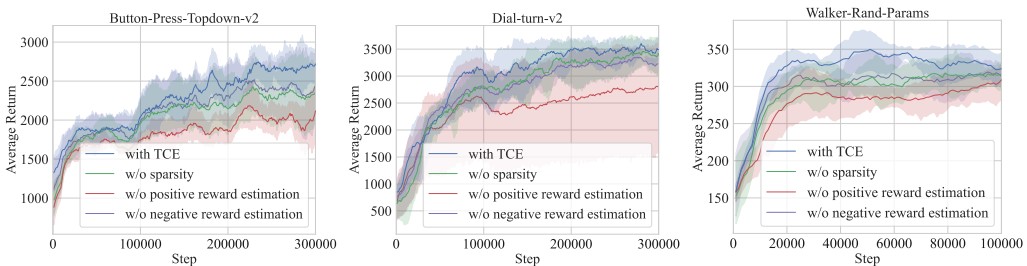

Figure 8: **Additional analysis of the effects of optimization perspectives on the task characteristic extractor.** Experiments in two task sets of Meta-World ML1 (Button-Press-Topdown and Dial-Turn), and the Walker-Rand-Params environment for exploring the effects of optimization perspectives on the task characteristic extractor when one of them is removed.

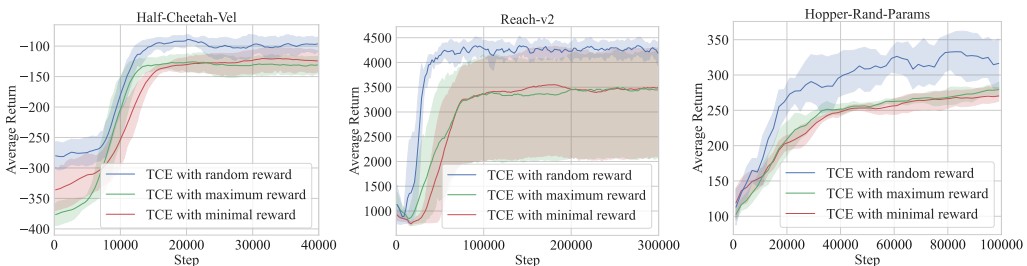

Figure 9: **Analysis of negative reward.** Experiments in the Half-Cheetah-Vel and Hopper-Rand-Params environments, and the Reach task set within Meta-World ML1 for analyzing the effects of different negative rewards. The variant named TCE with random rewards is the method we select for generating the negative reward $r'^{t\,neg}_i$. Meanwhile, the variants named TCE with the maximum reward and TCE with the minimum reward respectively mean methods that treat the maximum reward $r'^{max}_i$ or the minimum reward $r'^{min}_i$ from the trajectory $h'_i$ as negative rewards.

TCE with the minimum reward on TCMRL with only the task characteristic extractor. The results in Figure 9 depict that making the maximum or minimum reward the negative reward and employing them to optimize the task characteristic extractor yield similar but limited performance. This is due to the adverse effects of employing excessively biased negative rewards as constraints in the variants TCE with the maximum reward and TCE with the minimum reward. Moreover, in the Reach task set within Meta-World ML1, these two variants with excessively biased negative rewards (TCE with the maximum reward and TCE with the minimum reward) illustrate large performance fluctuations. In contrast, those negative rewards with appropriate random biases can effectively serve as constraints, aiding in optimizing the task characteristic extractor.

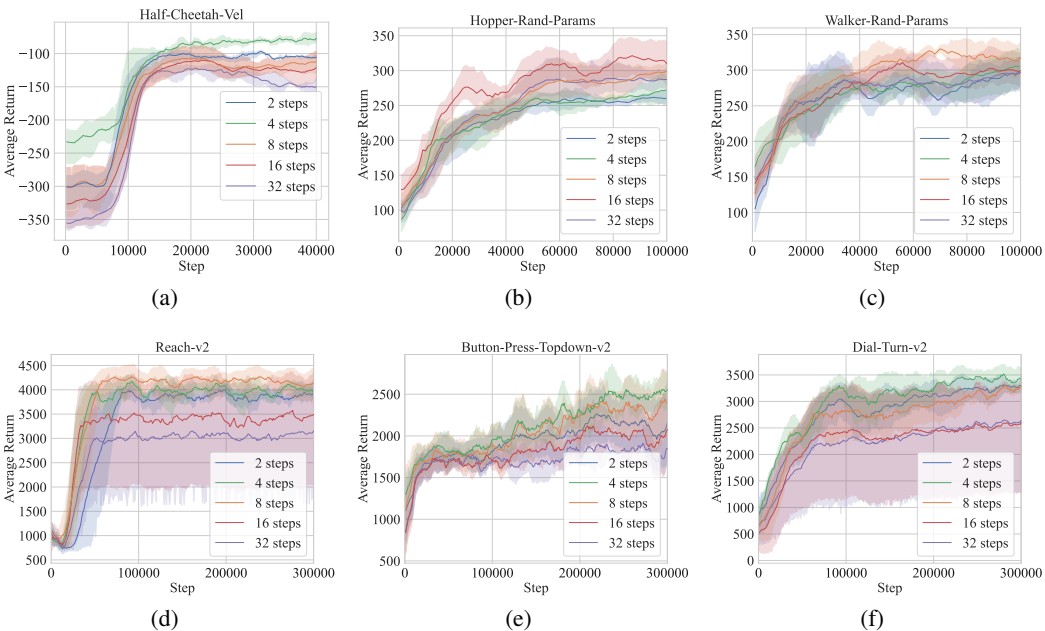

Figure 10: **Effect on length of subsequences.** Experiments in the Half-Cheetah-Vel, Hopper-Rand-Params and Walker-Rand-Params environments and three task sets within Meta-World ML1 (Reach, Button-Press-Topdown and Dial-Turn) for interrelations among transitions under various subsequence lengths, including 2, 4, 8, 16 and 32 steps.

### G.4 EFFECT ON LENGTH OF SUBSEQUENCES

We discover interrelations among transitions from $K$-step subsequences instead of the entire trajectory. Thus, $K$ is a crucial and fixed hyperparameter. Figure 10 shows our separate experiments to explore the impact of different values of $K \in \{2, 4, 8, 16, 32\}$ in three task sets of Meta-World ML1 (Button-Press-Topdown, Dial-Turn and Reach), and the Half-Cheetah-Vel, Hopper-Rand-Params and Walker-Rand-Params environments.

In the Half-Cheetah-Vel environment (see Figure 10(a)), when $K$ is set to 2, the performance and sample efficiency are suboptimal. This is because directly treating adjacent transition representations in a long sequence as subsequences results in incomplete interrelations among transitions. Although the interrelations discovered from such short subsequences are partially effective, they lack completeness. Yet, when $K$ is set to 4, the performance and sample efficiency are optimal. Discovering interrelations from subsequences of this length effectively enhances the adaptability of TCMRL to unseen target tasks. However, as we increase $K$ to larger values in $\{8, 16, 32\}$, the performance deteriorates significantly. This suggests that on longer subsequences, the limited interrelations among transition representations make it challenging to capture meaningful task contrastive information.

In the Hopper-Rand-Params environment (see Figure 10(b)), when $K$ is set to 2, the performance and sample efficiency are limited, due to the incomplete interrelations among transitions. When $K$ is set to 4, there is a performance improvement, but it remains suboptimal. This is because subsequences of length 4 are not suitable for discovering the interrelations among transitions. When $K$ is set to 8, the performance and sample efficiency are optimal, which means that 8 is an appropriate length for constructing subsequences. As in the case of the Half-Cheetah-Vel environment, as $K$ continues to increase, the performance instead keeps decreasing.

Overall, discovering interrelations from subsequences of appropriate length can indeed obtain overlooked interrelations among transitions, thereby capturing exhaustive task contrastive information. For different environments, achieving optimal performance and sample efficiency requires constructing subsequences with different values of $K$. For example, the optimal $K$ value is 8 in the Walker-Rand-Params environment and the Reach task set, while it is 16 in the Button-Press-Topdown and Dial-Turn task sets. Generally, as the value of $K$ increases, the performance initially improves until it reaches an appropriate value, and then declines. Such a trend is frequently observed, and there

may be special situations in certain environments. For instance, in the Reach and Dial-Turn task sets of Meta-World ML1, excessively high values of $K$ (16, 32) result in significant performance fluctuations. This is due to the challenge of effectively capturing meaningful task contrastive information from excessively long subsequences. Meanwhile, in the Walker-Rand-Params environment, the difference in performance between leveraging $K$ of 2 and 4 is minimal, since the limited interrelations among transitions are discovered from such relatively short subsequences.

### G.5 COMPARISON WITH FOCAL++

We compare TCMRL with FOCAL++ (Li et al., 2021a), which directly introduces the attention mechanism to achieve robust task inference. Specifically, FOCAL++ utilizes the attention mechanism from both sequence-wise attention and batch-wise attention perspectives. For each trajectory, the sequence-wise self-attention (Vaswani et al., 2017) is applied to capture the correlation within the transition dimensions. For each task, the batch-wise gated attention is applied to recalibrate the weights of transition samples. Although both TCMRL and FOCAL++ employ attention mechanisms, the implementation in TCMRL differs significantly. Specifically, the attention mechanism of the task characteristic extractor in TCMRL generates fine contexts based on the mean context encoding operation, and is optimized from the perspectives of positive and negative reward estimation with the context-based reward estimator and sparsity in attention weights. For a comprehensive comparison, we construct variants of FOCAL++ with these two distinct attention mechanisms, designated as FOCAL++(sequence-wise) and FOCAL++(batch-wise), and we compare these two variants with the original FOCAL++ and TCMRL in all experimental environments. The comparative results in Table 3 show that TCMRL outperforms FOCAL++, FOCAL++(sequence-wise) and FOCAL++(batch-wise) across most environments and task sets within Meta-World ML1, achieving effective adaptation to unseen target tasks.

### G.6 COMPARISON WITH CORRO

We compare TCMRL with CORRO (Yuan & Lu, 2022), which is a recent method that generates robust contexts through contrastive learning. Specifically, while treating different contexts corresponding to the same task as positive samples, it primarily constructs negative samples in two ways. First, in the cases where the overlap of state-action pairs between tasks is larger, it employs pre-trained condition variational auto-encoder (CVAE) (Sohn et al., 2015) for generating negative samples. Second, in the cases where the overlap of state-action pairs between tasks is small, it generates negative samples by reward randomization. The comparative results in Table 4 demonstrate that TCMRL outperforms both CORRO with CVAE and CORRO with RR across most environments and task sets within Meta-World ML1, showcasing superior performance.

### G.7 VISUALIZATION ANALYSIS

We report the t-SNE visualization (van der Maaten & Hinton, 2008) of the contexts of TCMRL, IDAQ, CSRO, CORRO, FOCAL++ and FOCAL in the Half-Cheetah-Vel environment in Figure 11, respectively. Additionally, we showcase two methods of generating negative samples in CORRO: CORRO with condition variational auto-encoder (CVAE) and CORRO with reward randomization (RR). In addition to FOCAL++, we also present two variants that correspond to the two attention mechanisms of FOCAL++: FOCAL++(sequence-wise) and FOCAL++(batch-wise). It provides a visual evidence of the efficacy of TCMRL in achieving effective clustering of context vectors. This observation showcases the capacity to effectively simultaneously preserve intrinsic similarity and extrinsic distinctiveness among corresponding contexts. However, in the visualizations corresponding to IDAQ, CSRO, FOCAL++, FOCAL++(sequence-wise), FOCAL++(batch-wise) and FOCAL, while the majority of contexts exhibit clustering effects, there are instances where the clustering of contexts for different tasks lacks tightness and leads to confusion. Specifically, the similarity among contexts belonging to the same task in IDAQ, CSRO and FOCAL is not sufficiently strong, and there is partial overlapping and ambiguity among contexts associated with different tasks. The visualization of contexts generated by CORRO with CVAE and CORRO with RR both indicate inadequate clustering trends. Local clustering is observed among different contexts corresponding to the same task, yet they exhibit a scattered distribution overall, while contexts corresponding to different tasks suffer from significant confusion.

Meanwhile, we present the t-SNE visualization in the Hopper-Rand-Params environment in Figure 12. The visualization results of contexts generated by TCMRL demonstrate a strong clustering tendency and clear boundaries among contexts of different tasks. It means that TCMRL can effectively capture the task characteristic information and task contrastive information. However, the visualization of contexts generated by IDAQ, CSRO and FOCAL++ exhibit good clustering properties, yet there are many

Table 3: Detailed comparison between FOCAL++ and TCMRL.

| Task set/Environment | TCMRL (ours) | FOCAL++ | FOCAL++(sequence-wise) | FOCAL++(batch-wise) |
|---|---|---|---|---|
| Assembly | **0.56±0.15** | 0.51±0.11 | 0.48±0.12 | 0.47±0.09 |
| Basketball | **0.82±0.11** | 0.71±0.25 | 0.68±0.17 | 0.66±0.09 |
| Bin-Picking | **0.65±0.10** | 0.51±0.24 | 0.39±0.26 | 0.46±0.14 |
| Box-Close | **0.62±0.09** | 0.44±0.03 | 0.42±0.03 | 0.40±0.06 |
| Button-Press-Topdown | **0.81±0.12** | 0.51±0.10 | 0.44±0.08 | 0.47±0.11 |
| Button-Press-Topdown-Wall | **0.47±0.02** | 0.42±0.02 | 0.39±0.02 | 0.38±0.04 |
| Button-Press | **0.81±0.05** | 0.79±0.05 | 0.76±0.09 | 0.75±0.09 |
| Button-Press-Wall | **1.07±0.03** | 0.98±0.07 | 0.90±0.08 | 0.88±0.14 |
| Coffee-Button | **0.83±0.12** | 0.75±0.16 | 0.71±0.13 | 0.65±0.19 |
| Coffee-Pull | **0.51±0.05** | 0.32±0.04 | 0.31±0.05 | 0.27±0.02 |
| Coffee-Push | **1.27±0.08** | 1.00±0.05 | 0.69±0.13 | 0.91±0.03 |
| Dial-Turn | **0.98±0.06** | 0.80±0.13 | 0.74±0.22 | 0.68±0.17 |
| Disassemble | **0.59±0.13** | 0.32±0.08 | 0.28±0.08 | 0.29±0.10 |
| Door-Close | **1.01±0.00** | 1.01±0.00 | 0.95±0.06 | 0.79±0.13 |
| Door-Lock | **0.99±0.00** | 0.96±0.00 | 0.96±0.00 | 0.95±0.01 |
| Door-Unlock | **1.18±0.02** | 1.11±0.02 | 1.09±0.01 | 1.06±0.01 |
| Door-Open | **1.00±0.00** | 0.92±0.01 | 0.92±0.00 | 0.87±0.03 |
| Drawer-Close | **1.01±0.01** | 0.97±0.01 | 0.96±0.06 | 0.96±0.04 |
| Drawer-Open | **0.90±0.03** | 0.84±0.05 | 0.71±0.14 | 0.73±0.02 |
| Faucet-Close | **1.13±0.01** | 1.11±0.00 | 1.10±0.01 | 1.10±0.01 |
| Faucet-Open | **1.08±0.02** | 1.06±0.00 | 1.05±0.01 | 1.00±0.05 |
| Hammer | **0.85±0.06** | 0.83±0.04 | 0.81±0.04 | 0.77±0.09 |
| Hand-Insert | **0.72±0.05** | 0.56±0.06 | 0.54±0.05 | 0.54±0.06 |
| Handle-Press-Side | **0.96±0.02** | 0.83±0.04 | 0.80±0.11 | 0.67±0.14 |
| Handle-Press | 0.77±0.06 | **0.90±0.02** | 0.89±0.01 | 0.89±0.01 |
| Handle-Pull-Side | **0.31±0.08** | 0.26±0.07 | 0.19±0.08 | 0.11±0.10 |
| Handle-Pull | **0.92±0.02** | 0.76±0.04 | 0.61±0.07 | 0.33±0.21 |
| Lever-Pull | **0.86±0.02** | 0.62±0.06 | 0.50±0.13 | 0.23±0.19 |
| Peg-Insert-Side | **0.45±0.05** | 0.19±0.07 | 0.14±0.07 | 0.12±0.05 |
| Peg-Unplug-Side | **0.69±0.01** | 0.50±0.03 | 0.45±0.04 | 0.40±0.06 |
| Pick-Out-Of-Hole | **0.71±0.06** | 0.29±0.17 | 0.18±0.20 | 0.27±0.18 |
| Pick-Place | **0.32±0.09** | 0.14±0.03 | 0.13±0.03 | 0.10±0.02 |
| Pick-Place-Wall | **0.43±0.15** | 0.18±0.06 | 0.16±0.07 | 0.16±0.03 |
| Plate-Slide-Back-Side | **0.97±0.03** | 0.96±0.02 | 0.93±0.02 | 0.92±0.03 |
| Plate-Slide-Back | **0.98±0.02** | 0.89±0.04 | 0.87±0.05 | 0.79±0.04 |
| Plate-Slide-Side | **1.07±0.08** | 0.99±0.07 | 0.96±0.07 | 0.88±0.11 |
| Plate-Slide | **1.01±0.02** | 0.92±0.01 | 0.90±0.01 | 0.86±0.03 |
| Push-Back | **0.58±0.04** | 0.21±0.05 | 0.20±0.07 | 0.18±0.05 |
| Push | **0.64±0.12** | 0.62±0.09 | 0.58±0.14 | 0.55±0.13 |
| Push-Wall | **0.77±0.11** | 0.74±0.07 | 0.71±0.09 | 0.55±0.03 |
| Reach | **0.92±0.06** | 0.87±0.04 | 0.81±0.04 | 0.69±0.07 |
| Reach-Wall | **0.94±0.05** | 0.92±0.03 | 0.92±0.03 | 0.83±0.02 |
| Shelf-Place | **0.84±0.12** | 0.53±0.04 | 0.37±0.11 | 0.34±0.05 |
| Soccer | **0.60±0.06** | 0.29±0.03 | 0.22±0.06 | 0.19±0.02 |
| Stick-Pull | **0.37±0.08** | 0.33±0.04 | 0.32±0.06 | 0.26±0.03 |
| Stick-Push | **0.85±0.05** | 0.83±0.05 | 0.76±0.08 | 0.69±0.05 |
| Sweep-Into | **0.66±0.02** | 0.58±0.03 | 0.53±0.05 | 0.47±0.03 |
| Sweep | **0.84±0.03** | 0.37±0.11 | 0.35±0.07 | 0.35±0.06 |
| Window-Close | **0.95±0.01** | 0.94±0.01 | 0.94±0.01 | 0.81±0.03 |
| Window-Open | **0.98±0.02** | 0.88±0.03 | 0.85±0.05 | 0.75±0.03 |
| Sparse-Point-Robot | **12.98±0.29** | 11.59±0.15 | 11.41±0.21 | 10.54±0.68 |
| Half-Cheetanh-Vel | **-79.7±11.3** | -116.7±14.9 | -124.2±10.1 | -151.9±28.4 |
| Point-Robot-Wind | **-4.75±0.26** | -4.89±0.31 | -4.91±0.29 | -6.22±0.39 |
| Hopper-Rand-Params | **368.62±10.37** | 318.86±20.14 | 299.41±36.16 | 291.79±44.72 |
| Walker-Rand-Params | **354.97±19.72** | 313.02±24.22 | 291.97±33.47 | 285.62±45.16 |

instances where boundaries between contexts of different tasks appear blurred or even overlapping. This suggests that they can effectively capture the task characteristic information, but struggle to obtain the task contrastive information. In the visualization result corresponding to FOCAL, contexts exhibit a certain degree of clustering tendency along with obvious confusion. It is extremely limited for FOCAL to capture both the task characteristic information and task contrastive information. The visualization of contexts generated by CORRO with CVAE, CORRO with RR, FOCAL++(sequence-wise) and FOCAL++(batch-wise) demonstrate poor clustering tendencies. Although there are instances of contexts within the same task clustering together locally, the boundaries between contexts corresponding to different tasks cannot be effectively delineated, resulting in significant confusion among contexts.

We also show the t-SNE visualization in the Reach task set within Meta-World ML1 in Figure 13. The visualization results of contexts corresponding to TCMRL exhibit clear clustering of contexts related to the same task, while those contexts of different tasks are separated. Nevertheless, the visualizations

Table 4: Detailed comparison between CORRO and TCMRL.

| Task set/Environment | TCMRL (ours) | CORRO with CVAE | CORRO with RR |
|---|---|---|---|
| Assembly | **0.56±0.15** | 0.34±0.08 | 0.38±0.08 |
| Basketball | **0.82±0.11** | 0.57±0.04 | 0.50±0.17 |
| Bin-Picking | **0.65±0.10** | 0.47±0.14 | 0.30±0.07 |
| Box-Close | **0.62±0.09** | 0.45±0.14 | 0.60±0.03 |
| Button-Press-Topdown | **0.81±0.12** | 0.55±0.14 | 0.21±0.24 |
| Button-Press-Topdown-Wall | **0.47±0.02** | 0.35±0.05 | 0.33±0.03 |
| Button-Press | **0.81±0.05** | 0.72±0.04 | 0.68±0.01 |
| Button-Press-Wall | **1.07±0.03** | 1.02±0.04 | 0.97±0.09 |
| Coffee-Button | **0.83±0.12** | 0.76±0.12 | 0.77±0.04 |
| Coffee-Pull | **0.51±0.05** | 0.43±0.04 | 0.22±0.06 |
| Coffee-Push | **1.27±0.08** | 1.14±0.09 | 1.17±0.16 |
| Dial-Turn | **0.98±0.06** | 0.87±0.07 | 0.83±0.02 |
| Disassemble | **0.59±0.13** | 0.24±0.12 | 0.49±0.06 |
| Door-Close | **1.01±0.00** | 0.84±0.12 | 0.98±0.01 |
| Door-Lock | **0.99±0.00** | 0.89±0.05 | 0.88±0.09 |
| Door-Unlock | **1.18±0.02** | 1.07±0.03 | 1.15±0.01 |
| Door-Open | **1.00±0.00** | 0.91±0.05 | 0.78±0.15 |
| Drawer-Close | **1.01±0.01** | 0.81±0.07 | 0.94±0.02 |
| Drawer-Open | **0.90±0.03** | 0.74±0.04 | 0.10±0.05 |
| Faucet-Close | **1.13±0.01** | 1.10±0.01 | 1.11±0.01 |
| Faucet-Open | **1.08±0.02** | 1.04±0.01 | 1.07±0.00 |
| Hammer | **0.85±0.06** | 0.79±0.05 | 0.64±0.09 |
| Hand-Insert | **0.72±0.05** | 0.57±0.10 | 0.63±0.13 |
| Handle-Press-Side | **0.96±0.02** | 0.33±0.10 | 0.28±0.16 |
| Handle-Press | **0.77±0.06** | 0.25±0.11 | 0.31±0.07 |
| Handle-Pull-Side | **0.31±0.08** | 0.10±0.06 | 0.08±0.09 |
| Handle-Pull | **0.92±0.02** | 0.63±0.19 | 0.33±0.15 |
| Lever-Pull | **0.86±0.02** | 0.78±0.06 | 0.81±0.03 |
| Peg-Insert-Side | **0.45±0.05** | 0.36±0.10 | 0.25±0.09 |
| Peg-Unplug-Side | **0.69±0.01** | 0.22±0.07 | 0.22±0.04 |
| Pick-Out-Of-Hole | **0.71±0.06** | 0.52±0.14 | 0.37±0.12 |
| Pick-Place | **0.32±0.09** | 0.15±0.05 | 0.25±0.05 |
| Pick-Place-Wall | **0.43±0.15** | 0.34±0.24 | 0.20±0.07 |
| Plate-Slide-Back-Side | **0.97±0.03** | 0.64±0.22 | 0.36±0.04 |
| Plate-Slide-Back | **0.98±0.02** | 0.80±0.04 | 0.21±0.05 |
| Plate-Slide-Side | **1.07±0.08** | 0.74±0.07 | 0.99±0.01 |
| Plate-Slide | **1.01±0.02** | 0.91±0.02 | 0.72±0.13 |
| Push-Back | **0.58±0.04** | 0.21±0.07 | 0.15±0.09 |
| Push | **0.64±0.12** | 0.57±0.08 | 0.56±0.06 |
| Push-Wall | **0.77±0.11** | 0.55±0.07 | 0.69±0.07 |
| Reach | **0.92±0.06** | 0.43±0.36 | 0.24±0.10 |
| Reach-Wall | **0.94±0.05** | 0.84±0.07 | 0.16±0.04 |
| Shelf-Place | **0.84±0.12** | 0.59±0.15 | 0.42±0.18 |
| Soccer | **0.60±0.06** | 0.54±0.02 | 0.58±0.07 |
| Stick-Pull | **0.37±0.08** | 0.33±0.04 | 0.24±0.08 |
| Stick-Push | **0.85±0.05** | 0.76±0.05 | 0.14±0.03 |
| Sweep-Into | **0.66±0.02** | 0.43±0.01 | 0.44±0.03 |
| Sweep | **0.84±0.03** | 0.53±0.18 | 0.22±0.22 |
| Window-Close | **0.95±0.01** | 0.92±0.01 | 0.89±0.01 |
| Window-Open | **0.98±0.02** | 0.48±0.02 | 0.46±0.01 |
| Half-Cheetanh-Vel | **-79.7±11.3** | -113.2±17.2 | -151.2±27.2 |
| Hopper-Rand-Params | **368.62±10.37** | 293.32±17.49 | 209.70±12.39 |
| Walker-Rand-Params | **354.97±19.72** | 301.49±5.06 | 295.60±12.44 |

of contexts generated by IDAQ and CSRO reveal only a partial clustering tendency, yet many contexts remain dispersed. The contexts of CSRO also exhibit instances of confusion. The visualization of contexts corresponding to FOCAL shows poor clustering tendencies, with contexts of only a few tasks achieving effective clustering. The visualizations of contexts generated by CORRO with CVAE and FOCAL++(batch-wise) demonstrate some degree of local clustering along with more pronounced confusion. Meanwhile, CORRO with RR and FOCAL++(sequence-wise) fail to generate contexts with task characteristic information and task contrastive information.

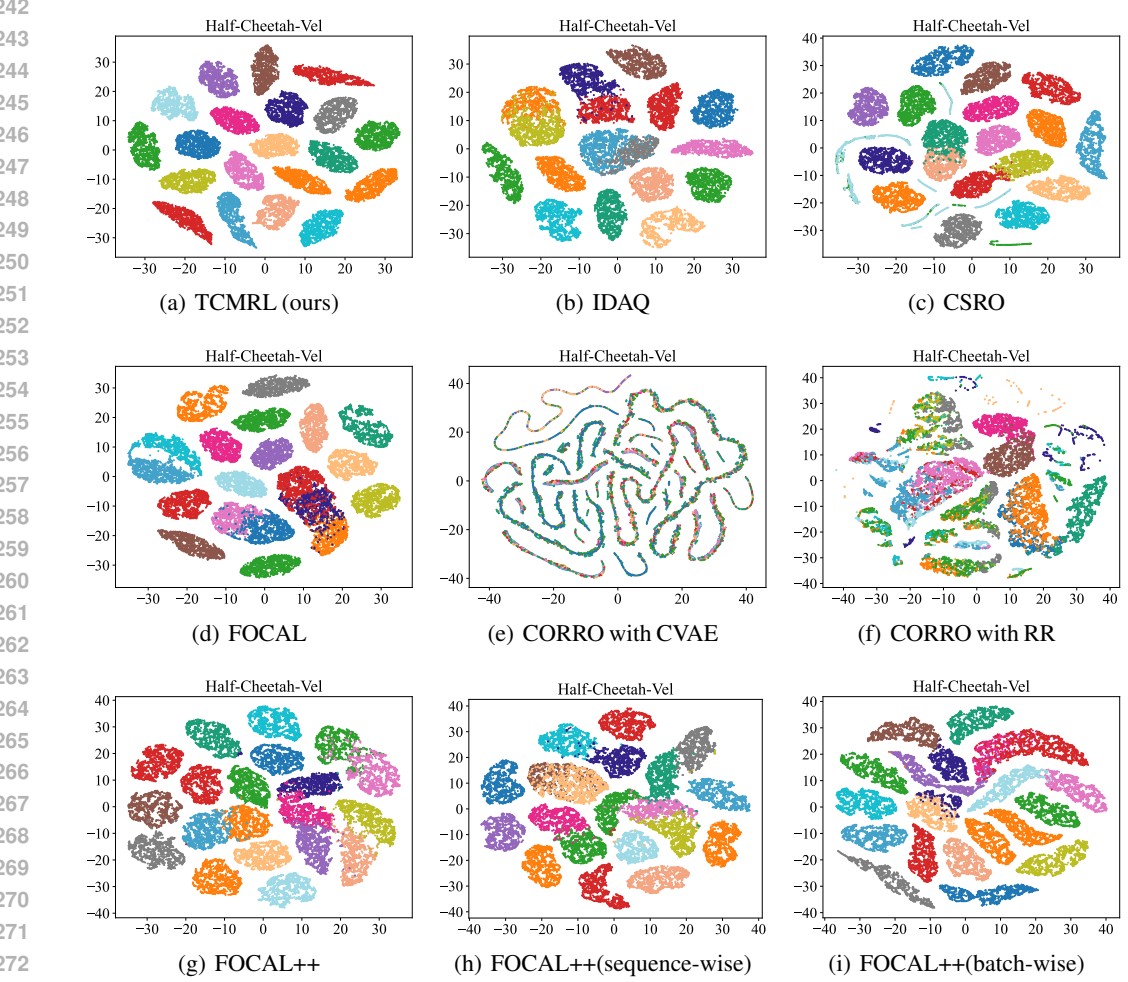

Figure 11: **t-SNE visualization in Half-Cheetah-Vel.** t-SNE visualization of the learned context vectors of TCMRL, IDAQ, CSRO, FOCAL, CORRO with CVAE, CORRO with RR, FOCAL++, FOCAL++(sequence-wise) and FOCAL++(batch-wise) drawn from 20 randomly selected tasks in the Half-Cheetah-Vel environment.

Overall, TCMRL can indeed provide a comprehensive understanding of tasks themselves and implicit relationships among tasks, resulting in generalizable contexts with both task characteristic information and task contrastive information.

### G.8 ANALYSIS IN REWARD-SPARSE ENVIRONMENT

We conduct experiments in the Spare-Point-Robot environment, which is reward-sparse. All tasks in this environment only differ by the reward function and conform to the following definition:

**Definition 1** (Reward-sparse transition). *For a particular task $\mathcal{T}_i$ within a reward-sparse environment, a transition of it $(s_i^t, a_i^t, s_i^{t+1}, R_i(s_i^t, a_i^t))$ is defined as a reward-sparse transition if $\forall \mathcal{T}_i \in \{\mathcal{T}\}, R_i(s_i^t, a_i^t) = c$. Following the policy invariance under reward transformations (Ng et al., 1999) and the setting in Li et al. (2021a), the constant c is assumed to be 0.*

**Definition 2** (Reward-sparse task). *For an offline dataset $\mathcal{D}_i = \{(s_i^t, a_i^t, s_i^{t+1}, R_i(s_i^t, a_i^t))\}$ corresponding to a particular task $\mathcal{T}_i$ within a reward-sparse environment, it includes two different types of transitions due to variations in rewards:*

$$\mathcal{D}_i = \{(s_i^t, a_i^t, s_i^{t+1}, R_i(s_i^t, a_i^t))\} \cup \{(s_i^t, a_i^t, s_i^{t+1}, 0)\}, \tag{19}$$

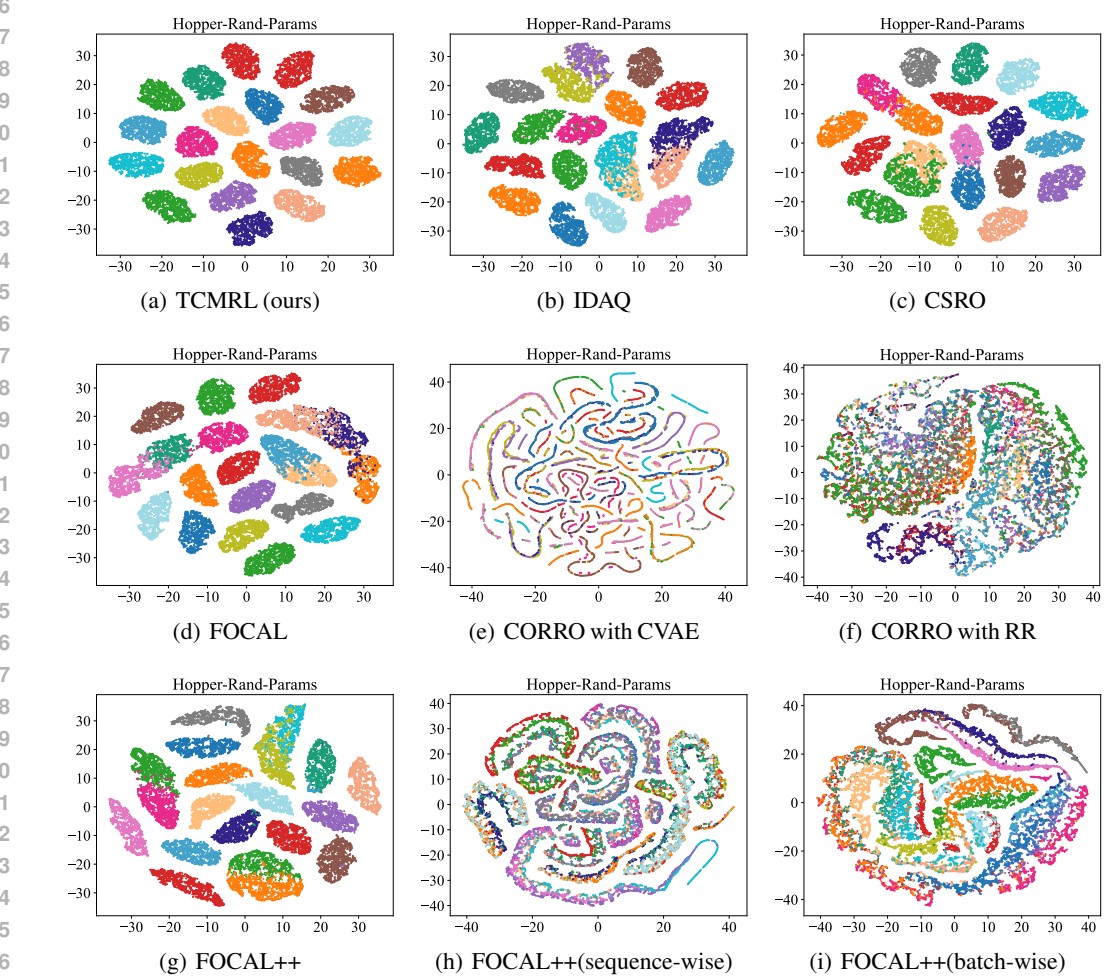

Figure 12: **t-SNE visualization in Hopper-Rand-Params.** t-SNE visualization of the learned context vectors of TCMRL, IDAQ, CSRO, FOCAL, CORRO with CVAE, CORRO with RR, FOCAL++, FOCAL++(sequence-wise) and FOCAL++(batch-wise), drawn from 20 randomly selected tasks in the Hopper-Rand-Params environment.

Table 5: Weights of transitions in the Spare-Point-Robot environment.

| Transitions | Mean of weights | Median of weights |
|---|---|---|
| Transitions with non-zero rewards | 0.0009839126 | 0.0009260265 |
| Transitions with zero rewards | 0.00081827847 | 0.0001407934 |

*where transitions $\{(s_i^t, a_i^t, s_i^{t+1}, 0)\}$ are reward-sparse transitions defined in Definition 1, while $\{(s_i^t, a_i^t, s_i^{t+1}, R_i(s_i^t, a_i^t))\}$ are the rest of the transitions. Additionally, the criteria used to categorize these transitions differ across various reward-sparse environments.*

Moreover, we propose a task characteristic extractor to identify transitions, within a trajectory, that are characteristic of a task, and assign high attention weights to these transitions when generating contexts. In reward-sparse environments, it should assign low attention weights to transitions with zero rewards. To evaluate its effectiveness, we conduct experiments in the Sparse-Point-Robot environment. Results presented in Table 5 indicate that although only a few transitions with non-zero rewards relate to the task characteristics, the attention weights for all transitions with non-zero rewards, both in terms of mean and median, are higher than those for transitions with zero rewards.

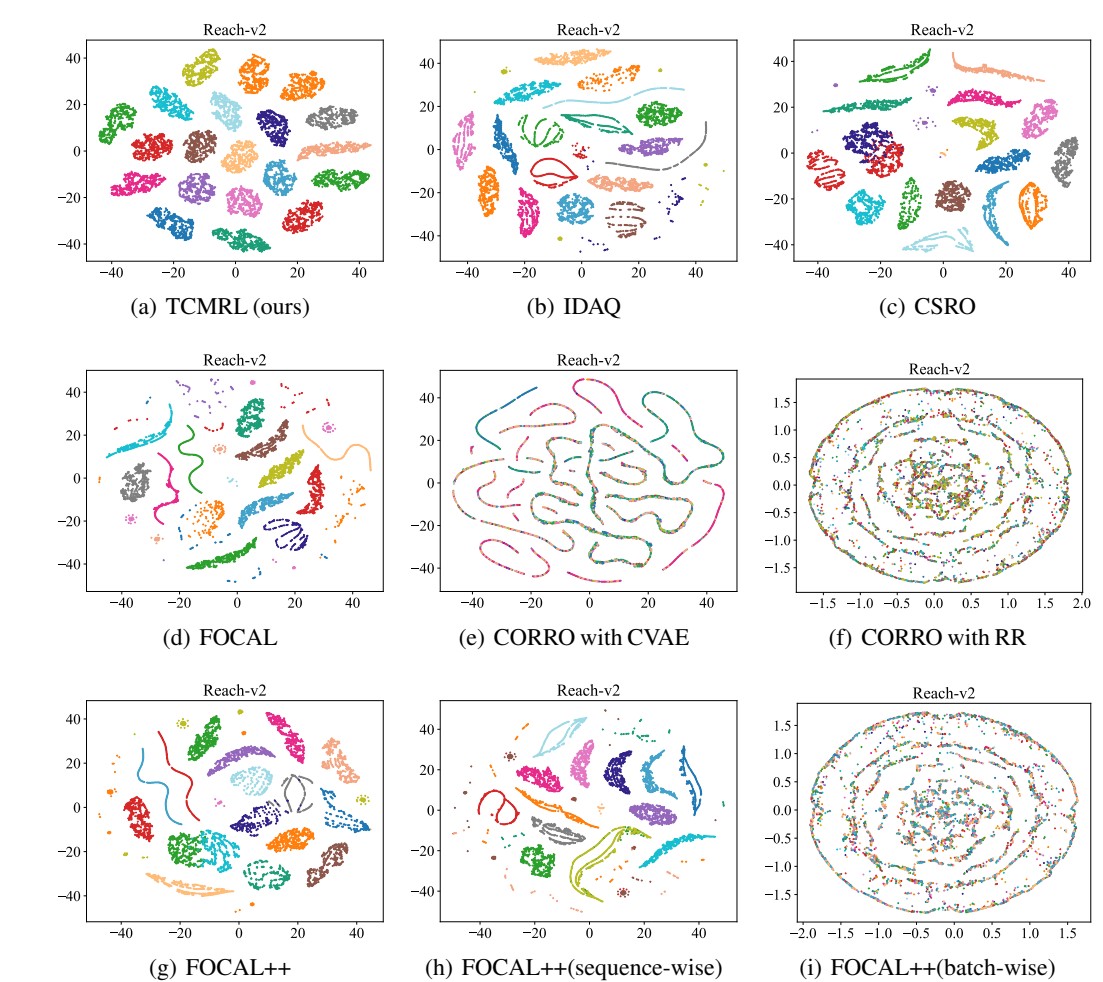

Figure 13: **t-SNE visualization in Reach-v2.** t-SNE visualization of the learned context vectors of TCMRL, IDAQ, CSRO, FOCAL, CORRO with CVAE, CORRO with RR, FOCAL++, FOCAL++(sequence-wise) and FOCAL++(batch-wise), drawn from 20 randomly selected tasks in the Reach task set within Meta-World ML1.

### G.9 COST ANALYSIS OF TCMRL

To access the computation costs of our proposed TCMRL framework, we experiment in the Half-Cheetah-Vel environment with an RTX 2080 Ti. Following the setting in Appendix F.2, our experiments consist of a total of 40000 steps. The results in Table 6 demonstrate that the computation costs of TCMRL are manageable and within accepted limits. Specifically, CORRO is not an end-to-end method, and the two-stage training process requires significantly more time. Moreover, to further analyze our task characteristic extractor and task contrastive loss, we calculate the computation costs for two variants: TCMRL w/o TCE (without task characteristic extractor) and TCMRL w/o TCL (without task contrastive loss). Results in Table 7 show that their computation costs are acceptable. Notably, due to the high and fluctuating GPU memory usage of the attention mechanism used by FOCAL++, we do not provide a corresponding analysis.

## H OFFLINE DATASET RETURNS

Table 8 shows the average returns of the offline datasets, which are utilized in the meta-training phase.

Table 6: Computation costs comparison between IDAQ, CSRO, CORRO and TCMRL.

| Method | Training time + Testing time | Training time | GPU memory usage |
|---|---|---|---|
| TCMRL (ours) | 4h26m10s | 2h10m29s | 1314MB |
| IDAQ | 4h34m24s | 1h51m40s | 1272MB |
| CSRO | 4h31m56s | 1h48m9s | 1274MB |
| CORRO | 6h31m51s | 3h56m8s | 1296MB |

Table 7: Computation costs analysis of TCE and TCL.

| Method | Training time + Testing time | Training time | GPU memory usage |
|---|---|---|---|
| TCMRL | 4h26m10s | 2h10m29s | 1314MB |
| TCMRL w/o TCE | 4h28m13s | 1h49m13s | 1294MB |
| TCMRL w/o TCL | 4h12m31s | 1h47m16s | 1272MB |

## I    POTENTIAL SOCIAL IMPACTS

The potential social impact includes the carbon footprint of the experiments and future work based on TCMRL.

Table 8: Dataset average returns on experimental environments.

| Task/Environment | Dataset return |
| --- | --- |
| Assembly | 4275.42 |
| Basketball | 4086.65 |
| Bin-Picking | 4254.52 |
| Box-Close | 4009.24 |
| Button-Press-Topdown | 3563.66 |
| Button-Press-Topdown-Wall | 3774.26 |
| Button-Press | 3857.22 |
| Button-Press-Wall | 2886.57 |
| Coffee-Button | 4035.74 |
| Coffee-Pull | 4205.87 |
| Coffee-Push | 1531.27 |
| Dial-Turn | 3840.01 |
| Disassemble | 3940.74 |
| Door-Close | 4487.84 |
| Door-Lock | 3352.69 |
| Door-Unlock | 3585.54 |
| Door-Open | 4455.43 |
| Drawer-Close | 4238.61 |
| Drawer-Open | 4045.54 |
| Faucet-Close | 4033.40 |
| Faucet-Open | 4147.91 |
| Hammer | 4274.10 |
| Hand-Insert | 3744.95 |
| Handle-Press-Side | 4969.13 |
| Handle-Press | 4794.29 |
| Handle-Pull-Side | 2838.50 |
| Handle-Pull | 3907.89 |
| Lever-Pull | 923.90 |
| Peg-Insert-Side | 3797.08 |
| Peg-Unplug-Side | 4128.75 |
| Pick-Out-Of-Hole | 3573.91 |
| Pick-Place | 3560.12 |
| Pick-Place-Wall | 2437.79 |
| Plate-Slide-Back-Side | 4721.63 |
| Plate-Slide-Back | 4726.75 |
| Plate-Slide-Side | 3517.98 |
| Plate-Slide | 4390.86 |
| Push-Back | 3809.79 |
| Push | 3016.54 |
| Push-Wall | 3721.30 |
| Reach | 4873.55 |
| Reach-Wall | 4804.65 |
| Shelf-Place | 2802.44 |
| Soccer | 2841.48 |
| Stick-Pull | 4147.37 |
| Stick-Push | 4124.41 |
| Sweep-Into | 4061.64 |
| Sweep | 4356.22 |
| Window-Close | 3583.68 |
| Window-Open | 4320.14 |
| Sparse-Point-Robot | 7.24 |
| Half-Cheetanh-Vel | -138.29 |
| Point-Robot-Wind | -7.84 |
| Hopper-Rand-Params | 450.84 |
| Walker-Rand-Params | 496.33 |

