# OpenReview forum: "Task Characteristic and Contrastive Contexts for Improving Generalization in Offline Meta-Reinforcement Learning"
_ICLR.cc/2025/Conference — Submitted to ICLR 2025_

### Official Review · Reviewer_SQK4 · 2024-10-31

**Soundness:** 3
**Presentation:** 3
**Contribution:** 3
**Rating:** 6
**Confidence:** 4

**Summary:**

The paper addresses the limitations in generalization and adaptation of existing context-based offline meta-reinforcement learning (meta-RL) methods. The proposed framework, TCMRL, enhances context generation by incorporating both task characteristic information, which identifies key transitions within tasks, and task contrastive information, which distinguishes tasks through interrelations in trajectory subsequences. This combined approach yields a comprehensive task understanding, improving adaptability to unseen tasks. Experiments confirm TCMRL’s advantage in generating generalizable contexts and effective adaptation over previous methods.

**Strengths:**

•   Clarity: The paper is well-articulated, with a clear and complete structure that makes the methodology and findings accessible to readers. Additionally, the appendix provides extensive experimental details and analyses.
•   Significance: I believe the authors have targeted a valuable goal, namely addressing the challenge of context shift in context-based offline meta-reinforcement learning methods. The paper demonstrates consistent improvements over baseline methods across a wide range of experiments, showcasing the robustness of their approach in tackling this important issue.

**Weaknesses:**

Quality: The paper lacks a strong motivational foundation, particularly in explaining why task characteristic information and task contrastive information are expected to enhance context generalization. While the authors introduce a novel method based on these two types of information, the construction of the approach appears somewhat arbitrary, relying on intuition rather than solid theoretical underpinnings. An improved presentation could include theoretical justifications or empirical evidence demonstrating that capturing these specific forms of task information is indeed crucial for generalization.
•   Originality: Although the technical implementation is undoubtedly innovative in its details, the underlying concepts are relatively familiar within the field. Techniques such as implicit attention mechanisms, context encoding, and task-based contrastive learning have been explored previously, and this paper can be seen as a new combination of these existing ideas.

**Questions:**

1.	As illustrated in the weaknesses, why is the lack of generalization attributed to the absence of task characteristic information and task contrastive information? Could you explain this in more detail, and how do you perceive the relationship between these two types of information?
2.	When extracting task characteristic information, why not consider using a well-established architecture like the Transformer? Given that Transformers leverage self-attention mechanisms to extract key information from sequences and create unified representations while capturing internal relationships within sequences, it seems like a viable option.
3.	Could you provide the rationale for designing the negative reward estimation as you did? What motivated this specific design?
4.	How do you determine the proportions of the various losses in the optimization process? I believe that the hyperparameters setting these ratios significantly impact the method’s performance.
5.	In the ablation study, as shown in Figures 5 and 6, I noticed that, in experiments like reacher-v2, removing an individual component within TCE results in greater performance loss than fully removing TCE. How would you explain this phenomenon?

---

> ### Author Response · Authors · 2024-11-18
> **Response to question (1) - (3)**
>
> Thank you for taking the time to review our paper and for providing valuable feedback.
>
> ``Q1. As illustrated in the weaknesses, why is the lack of generalization attributed to the absence of task characteristic information and task contrastive information? Could you explain this in more detail, and how do you perceive the relationship between these two types of information?``
>
> Our method is based on both empirical evidence and existing theoretical foundations. We have introduced the concept of context shift (see Lines 39-45, Section 1), the limitations of existing context-based offline meta-RL methods (FOCAL (Li et al., 2021b), FOCAL++ (Li et al., 2021a), CORRO (Yuan & Lu, 2022), IDAQ (Wang et al., 2023) and CSRO (Gao et al., 2023)) (see Lines 136-144, Section 2), the motivation of the task characteristic extractor (see Lines 207-217, Section 4.1.1) and the motivation of the task contrastive loss (Lines 307-317, Section 4.1.2).
>
> Based on these previous statements, we will further analyze TCMRL in detail. When learning from offline datasets of meta-training tasks, existing context-based offline meta-RL methods fail to generate contexts that exhibit consistency within the same task and distinctness among different tasks, reflecting comprehensive internal relationships among tasks. We experimentally demonstrate that these properties correspond to task characteristic information and task contrastive information, respectively. TCMRL aims to capture both of them to generate generalizable contexts with exhaustive task information and construct comprehensive internal relationships among tasks. These relationships enable the extension of knowledge from offline datasets of meta-training tasks to unseen target tasks, facilitating efficient and effective adaptation to unseen target tasks.
>
> These two types of information are complementary components of task information, and their roles are described in Lines 49-50, Section 1 and Lines 81-82, Section 1. By combining them, we can obtain complete task information, which allows us to build comprehensive internal relationships among tasks and generate generable contexts.
>
> ``Q2. When extracting task characteristic information, why not consider using a well-established architecture like the Transformer? Given that Transformers leverage self-attention mechanisms to extract key information from sequences and create unified representations while capturing internal relationships within sequences, it seems like a viable option.``
>
> Thanks for your comment. Actually, we have already considered using structures such as the Transformer before. However, as mentioned in Line 1397, Section G.9, compared to our attention mechanism based on MLP, they result in high and fluctuating GPU memory usage, as well as increased time costs, without providing significant performance improvements.
>
> To further validate this observation, we have conducted additional experiments with the Transformer in the Half-Cheetah-Vel, Hopper-Rand-Params, and Walker-Rand-Params environments, as well as three task sets from Meta-World ML1 (Button-Press-Topdown, Dial-Turn, and Reach). Notably, when using Transformer, our L_{TCE}^{spa} (Eq. (6)) is not utilized. The results are shown as follows:
>
> |                      | TCMRL            | TCMRL with Transformer |
> | -------------------- | ---------------- | ---------------------- |
> | Half-Cheetah-Vel     | **-79.7±11.3**   | -95.53±17.4            |
> | Hopper-Rand-Params   | **368.62±10.37** | 337.40±21.05           |
> | Walker-Rand-Params   | **354.97±19.72** | 335.29±27.16           |
> | Button-Press-Topdown | **0.81±0.12**    | 0.56±0.08              |
> | Dial-Turn            | **0.98±0.01**    | 0.84±0.07              |
> | Reach                | **0.92±0.03**    | 0.88±0.05              |
>
> ``Q3. Could you provide the rationale for designing the negative reward estimation as you did? What motivated this specific design?``
>
> As mentioned in Lines 283-287, Section 4.1.1, we empirically validate that for optimizing our task characteristic extractor solely from the perspective of positive reward estimation, which focuses on identifying and emphasizing transitions that are task characteristics, its effect is limited. For this reason, we propose the novel idea of negative reward estimation. By inducing wrong estimations through c_{i}^{neg}, which primarily captures the redundant information of less important transitions, we indirectly optimize the task characteristic extractor.
>
> If important transitions are contained in the remaining transitions, they tend to guide the estimation towards the correct target {r’}_{i}^{t} instead of the wrong target {{r'}_{i}^{t}}^{neg}.

---

> ### Author Response · Authors · 2024-11-18
> **Response to question (4) - (5)**
>
> Thank you for taking the time to review our paper and for providing valuable feedback.
>
> ``Q4. How do you determine the proportions of the various losses in the optimization process? I believe that the hyperparameters setting these ratios significantly impact the method’s performance.``
>
> We use the popular grid search to determine the hyperparameters. As shown in Figures 6, 7 and 8, the perspective of positive reward estimation plays a major role, and therefore we assign it a higher proportion. For simplicity, we generally assign equal proportions to the perspectives of sparsity and negative reward estimation, which are treated as constraint terms. Additionally, we use the same hyperparameter settings in most task sets within the Meta-World ML1 environment.
>
>
> ``Q5. In the ablation study, as shown in Figures 5 and 6, I noticed that, in experiments like reacher-v2, removing an individual component within TCE results in greater performance loss than fully removing TCE. How would you explain this phenomenon?``
>
> Thanks for your detailed review. As stated in Line 475, Section 5.4, the results in Figure 6 are obtained without using the task contrastive loss, so they cannot be directly compared with the "w/o TCE" results in Figure 5. We acknowledge that there is ambiguity in the legend of Figure 6, and we will clarify it in the subsequent version.
>
> Additionally, as discussed in Lines 480-525, Section 5.4, during the optimization of our task characteristic extractor, the perspective of positive reward estimation plays a major role, while the perspectives of sparsity in attention weights and negative reward estimation serve as constraints. Consequently, when optimization is performed without the perspective of positive reward estimation, it may lead to a significant performance decline.

---

### Official Review · Reviewer_QaPj · 2024-11-04

**Soundness:** 2
**Presentation:** 1
**Contribution:** 2
**Rating:** 3
**Confidence:** 4

**Summary:**

This work targets the problem of offline meta RL by learning a context of a task information from trajectories so that the learned context encoder can quickly capture characteristics of an unseen test task with limited interactions. Specifically, the authors propose to learn such context encoder by conditioning a reward neural network on a weighted aggregation of transition encodings in a trajectory. The authors also propose to train the context vector by penalising rewards prediction when conditioned on a reversed weighted version of context. This work also leverages contrastive learning to train transition encoding.

**Strengths:**

The motivation behind learning both task characteristic and task contrastive information for better meta generalisation is reasonable.

The proposed method is evaluated on many meta RL environments and empirical results show improved performance.

**Weaknesses:**

The proposed method in this work consists of many components around the context encoder training. However, It is unclear to me what is the fundamental technical reason behind these kinds of design and why these specific designs can achieve desired behaviour of the context encoder. There are many explanations in the method part in section 4 but they are not well structured in logic and look very lengthy:

(1) Line 38: “as only a few key transitions within the trajectory provide the main task characteristic information…” This is to say many other transitions do not distinguish tasks. I have concern over this statement as this is only probably correct when the tasks have some property like a hierarchical structure. In general when the dynamics of a target task has a consistent shift on the entire state space, such sparsity prior would not be beneficial.

(2) It is unclear to me why Eq. (7) and Eq. (8) can lead to learning a context encoder such that the task characteristic extractor q can capture task unique transitions. The neural network is probably able to capture task conditioned reward with a task-level context without learning relations in terms of tasks transitions. In my opinion, the network does not promote the correct importance score of c_i. It probably only makes c_i and c_i^neg different and that is enough to learn a conditional reward function under Eq. (7) and (8).

(3) Are r and r_reverse in Eq. (7) and Eq. (8) the same neural network with same parameters?

(4) It seems that Eq. (7) and Eq. (8) only capture the task shift in terms of reward function while the transition dynamics is ignored (no loss function in terms of next state prediction). Can authors please explain the reason?

Overall, the proposed method consists of several modified versions of previous loss functions and is also combined with existing contrastive learning technique. The technical novelty is not strong and there is no theoretical analysis on why the proposed objective function can guarantee generalisation in a meta learning setting.

**Questions:**

Please see the weakness part.

---

> ### Author Response · Authors · 2024-11-18
> **Response to the weakness about fundamental technical reason and weakness (1)**
>
> Thank you for taking the time to review our paper and for providing valuable feedback.
>
> ``Weakness about fundamental technical reason. The proposed method in this work consists of many components around the context encoder training. However, It is unclear to me what is the fundamental technical reason behind these kinds of design and why these specific designs can achieve desired behaviour of the context encoder. There are many explanations in the method part in section 4 but they are not well structured in logic and look very lengthy.``
>
> We believe there may be some misunderstandings. TCMRL is proposed based on both empirical evidence and existing theoretical foundations and built upon a clear high-level motivation. We have introduced the motivation of the task characteristic extractor (see Lines 207-217, Section 4.1.1) and the motivation of the task contrastive loss (Lines 307-317, Section 4.1.2).
>
> In addition to these statements mentioned in our paper, we will further explain the overall motivation behind TCMRL. In cases where existing context-based offline meta-RL methods (FOCAL (Li et al., 2021b), FOCAL++ (Li et al., 2021a), CORRO (Yuan & Lu, 2022), IDAQ (Wang et al., 2023) and CSRO (Gao et al., 2023)) do not consider constructing comprehensive internal relationships among tasks from offline data of meta-training tasks, TCMRL aims to reduce the Impact of context shift (see Lines 39~45, Section 1) by leveraging these relationships. These relationships are directly reflected in the contexts of the same task and the contexts of different tasks. Our goal is to capture task characteristic information and task contrastive information to generate contexts that exhibit both consistency within the same task and distinctness among different tasks. This allows us to construct comprehensive internal relationships among tasks, enabling the knowledge from offline datasets of meta-training tasks to be extended to unseen target tasks based on these relationships.
>
>
>
> ``W1. Line 238: “as only a few key transitions within the trajectory provide the main task characteristic information…” This is to say many other transitions do not distinguish tasks. I have concern over this statement as this is only probably correct when the tasks have some property like a hierarchical structure. In general when the dynamics of a target task has a consistent shift on the entire state space, such sparsity prior would not be beneficial.``
>
> We do not aim to use task characteristic information to distinguish different tasks. As mentioned in Lines 49-50, Section 1, task characteristic information reflects the consistency of contexts within the same task. This statement is supported by the theoretical foundations provided in two references (Arjona-Medina et al., 2019; Faccio et al., 2022) cited in Line 239. Our goal is to extract stable contexts for a particular task from its different trajectories by identifying and emphasizing transitions that are task characteristics, rather than being affected by the redundant information from less important transitions. The contexts that encompass comprehensive task information will directly guide the context-based policy to complete the corresponding task. Therefore, identifying and emphasizing transitions that are task characteristics within different trajectories of the same task is crucial.
>
> Moreover, as mentioned in Lines 81-82, we aim to distinguish tasks from one another by capturing task contrastive information. We design a task contrastive loss that discovers overlooked interrelations among transitions from trajectory subsequences through contrastive learning for capturing exhaustive task contrastive information (see Section 4.1.2).
>
> We acknowledge that situations where "the dynamics of a target task has a consistent shift on the entire state space" do exist. Currently, neither the existing context-based offline meta-RL methods nor TCMRL can adequately address such challenging scenarios. In fact, the performance we report is not based on a single unseen target task but represents the average performance across a series of unseen target tasks. Some tasks in this series may fall into this category and exhibit low performance. We consider this limitation as one of the directions for future research.

---

> ### Author Response · Authors · 2024-11-18
> **Response to weakness (2) - (4)**
>
> Thank you for taking the time to review our paper and for providing valuable feedback.
>
> ``W2. It is unclear to me why Eq. (7) and Eq. (8) can lead to learning a context encoder such that the task characteristic extractor q can capture task unique transitions. The neural network is probably able to capture task conditioned reward with a task-level context without learning relations in terms of tasks transitions. In my opinion, the network does not promote the correct importance score of c_i. It probably only makes c_i and c_i^neg different and that is enough to learn a conditional reward function under Eq. (7) and (8).``
>
> In Eq. (7), we use the context-based reward estimator as the metric to ensure that our task characteristic extractor can still effectively express task information, even when it assigns high attention weights to only a few transitions. This helps guide the task characteristic extractor to identify and emphasize transitions that represent the task characteristics (see Lines 268-269, Section 4.1.1).
>
> In Eq. (8), we assign high attention weights to the remaining transitions and induce wrong reward estimations in the context-based reward estimator. This further mitigates the impact of redundant information from these transitions and indirectly emphasizes task unique transitions (see Lines 283-287, Section 4.1.1).
>
> In summary, in Eq. (7), if less important transitions are assigned high attention weights, accurate reward estimations become difficult. Meanwhile, in Eq. (8), if important transitions are treated as part of the remaining transitions, the reward estimations are less likely to converge toward wrong targets. Together, these effects enable the optimization of our task characteristic extractor.
>
> ``W3. Are r and r_reverse in Eq. (7) and Eq. (8) the same neural network with same parameters?``
>
> Yes, your understanding is right. Both \hat{r} and {\hat{r}}_{reverse} are computed by a same neural network with the same parameters.
>
> ``W4. It seems that Eq. (7) and Eq. (8) only capture the task shift in terms of reward function while the transition dynamics is ignored (no loss function in terms of next state prediction). Can authors please explain the reason?``
>
> We believe there may be a misunderstanding. We have considered constructing the context-based dynamic estimator in the same way. However, since both the context-based dynamic estimator and our context-based reward estimator serve as ways to measure how well the context captures task characteristic information (Lines 249-250, Section 4.1.1), we have chosen to introduce only the context-based reward estimator based on our experimental results. Our experimental results show that adding the context-based dynamic estimator does not lead to significant performance improvement but instead increases computational cost.
>
> To further validate this observation, we have conducted additional experiments in the Half-Cheetah-Vel, Hopper-Rand-Params, and Walker-Rand-Params environments, as well as three task sets from Meta-World ML1 (Button-Press-Topdown, Dial-Turn, and Reach).
>
> The experimental results show that the performance results of TCMRL with reward estimations, TCMRL with dynamic estimations, and TCMRL with both reward and dynamic estimations are similar, with only slight variations that can be attributed to error fluctuations. Notably, to construct the learning objective for negative dynamic estimation, we sample r^{noise} in the same way from a multidimensional Gaussian distribution of noise to build {{s'}_{i}^{t+1}}^{neg}. The results are shown as follows:
> |                      | TCMRL with reward estimations (ours) | TCMRL with dynamic estimations | TCMRL with reward and dynamic estimations |
> | -------------------- | ------------------------------------ | ------------------------------ | ----------------------------------------- |
> | Half-Cheetah-Vel     | -79.7±11.3                           | -81.9±12.6                     | **-79.1±14.3**                            |
> | Hopper-Rand-Params   | **368.62±10.37**                     | 358.33±15.43                   | 366.95±12.50                              |
> | Walker-Rand-Params   | **354.97±19.72**                     | 347.40±23.18                   | 351.06±17.01                              |
> | Button-Press-Topdown | **0.81±0.12**                        | 0.80±0.11                      | 0.81±0.11                                 |
> | Dial-Turn            | **0.98±0.01**                        | 0.98±0.01                      | 0.98±0.01                                 |
> | Reach                | **0.92±0.03**                        | 0.90±0.03                      | 0.92±0.02                                 |

---

> > ### Author Response · Authors · 2024-11-18
> > **Response to the weakness about novelty**
> >
> > ``Weakness about novelty. Overall, the proposed method consists of several modified versions of previous loss functions and is also combined with existing contrastive learning technique. The technical novelty is not strong and there is no theoretical analysis on why the proposed objective function can guarantee generalisation in a meta learning setting.``
> >
> > Thank you for taking the time to review our paper and for providing valuable feedback. Although the attention mechanism, loss functions, and the contrastive learning technique we utilize have their roots in other domains, we believe that TCMRL is novel to the context-based offline meta-RL setting for the following reasons:
> >
> > **(1) We propose a task characteristic extractor that builds upon the coarse contexts obtained through averaging in existing context-based offline meta-RL methods. It identifies and emphasizes transitions that represent task characteristics. We introduce the context-based reward estimator as a way to measure and to optimize the task characteristic extractor from the perspectives of positive and negative reward estimation and sparsity in attention weights.**
> >
> > **(2) We discover overlooked interrelations among transitions from trajectory subsequences. These interrelations are modeled as the mutual information among transitions within each subtrajectory, and the lower bound of the mutual information is approximated using the InfoNCE loss function.**
> >
> > **(3) We construct a novel combination of (1) and (2) and specifically adapt it for the context-based offline meta-RL setting. TCMRL captures both task characteristic information and task contrastive information to improve the generalization of contexts.**

---

> > > ### Author Response · Authors · 2024-11-25
> > > **Gentle Reminder**
> > >
> > > Dear Reviewer QaPj
> > >
> > > We would like to know if our response has addressed your concerns and questions. If you have any further concerns or suggestions for the paper or our rebuttal, please let us know. We would be happy to engage in further discussion and manuscript improvement. Thank you again for the time and effort you dedicated to reviewing this work.

---

> ### Comment · Reviewer_QaPj · 2024-11-25
> **Response to the authors**
>
> I would like to thanks the authors for their detailed responses.
>
> Regarding the “Weakness about fundamental technical reason.”: I agree with the authors that this work has a clear high-level motivation. This is also one of the strengths I have mentioned in the review. However, this reasonable overall motivation does not fully support the *technical* reason behind each design choice of many loss functions proposed in the technical part, i.e., these high-level goals do not automatically translate to these loss functions. The proposed loss function might improve the model in terms of these goals, but the linkage here is not strong. This concern of intuitive modelling is also shared with reviewer SQK4.
>
> Regarding W1, I would like to thank the authors for their further explanations. It seems that consistent shift is not that easy and offline meta-RL methods can be limited in this setting, but overall I think this is not a very big problem during this phase. I would suggest the authors improve the wording in this paragraph (Line 238) by pointing out what the potential successful or failure cases are for this design. For W2, the explanations do not solve my concern. It is still unclear to me why these two loss functions can learn correct attention. I can understand that learning with Eq. (7) or (8) alone obviously fails but the response did not answer whether simply making c_i and c_i^neg different is enough for the minimization of these two loss function. “in Eq. (7), if less important transitions are assigned high attention weights, accurate reward estimations become difficult. Meanwhile, in Eq. (8), if important transitions …” are there any theoretical or empirical justifications for these two points in this work? For W4, I would like to thank the authors for results of comparison. It seems that these results do not provide any insights regarding different choices and it might be interesting to investigate the reason behind this.

---

> > ### Author Response · Authors · 2024-11-26
> > **Thanks for your response**
> >
> > Thank you for your response. We appreciate the discussion!
> >
> > Thank you for your recognition and appreciation of our response to W1. Our sparsity prior setup is common in other fields, especially in data-driven approaches [1, 2, 3, 4]. We will improve our presentation and add relevant potential successful or failure cases in the subsequent version.
> >
> > **Regarding the "Weakness about fundamental technical reason":**
> >
> > To optimize our task characteristic extractor, we propose loss functions from three perspectives: positive and negative reward estimation, and sparsity in attention weights. These loss functions are grounded in our sparsity prior, which is supported by theoretical foundations (Arjona-Medina et al., 2019; Faccio et al., 2022).
> >
> > As mentioned in Lines 240-245, Section 4.1.1, the motivation for our L_{TCE}^{spa} is to assign higher attention weights to transitions that represent task characteristics, rather than to general transitions containing redundant information.
> >
> > The motivation for our L_{TCE}^{pos} is to guide our task characteristic extractor to identify and emphasize only a few transitions that are task characteristics, enabling contexts that comprehensively reflect the task characteristic information of a particular task. This loss function ensures c_i achieves accurate reward estimation for both important and general transitions, forcing the task characteristic extractor to focus on task characteristic transitions.
> >
> > The motivation for our L_{TCE}^{neg} is to optimize the task characteristic extractor from the opposite perspective. Our primary goal is to ensure that c_{i}^{neg}, which focuses on redundant information from general transitions, fails to achieve accurate reward estimation, thereby guiding the task characteristic extractor to assign low attention weights to such transitions. To achieve this, as described in Lines 277-291, Section 4.1.1, we design a negative target {r'}_{i}^{neg} and induce the reward estimation conditioned on c_{i}^{neg} to approximate this incorrect target.
> >
> > **Regarding  W2:**
> >
> > In Eq. (7) and Eq. (8), two distinct reward estimation targets are used: {r'}_{i}^{t}, the positive target sampled from D_i, and {{r'}_{i}^{t}}^{neg}, the negative target. For a given task, {r'}_{i}^{t} is related to the reward function that reflects the task information, while {{r'}_{i}^{t}}^{neg} introduces bias. Eq. (7) encourages the task characteristic extractor to assign high attention weights to task characteristic transitions for achieving accurate reward estimation. Eq. (8) enforces the task characteristic extractor to assign low attention weights to general transitions for approximating the negative target. Due to the differences in task characteristic information represented by these two types of transitions, incorrect assignment of attention weights disrupts the optimization, preventing the proper minimization of both Eq. (7) and Eq. (8). Some prior studies [1, 3] have demonstrated the empirical effectiveness of optimizing in both positive and negative perspectives, providing justification for our method. Moreover, as shown in the ablation results in Figure 6, Figure 7, and Figure 8, L_{TCE}^{pos} plays a primary role, while L_{TCE}^{neg} serves as a constraint to assist the optimization process. Notably, as stated in Lines 299-301, Section 4.1.1, L_{TCE}^{neg} is not used to optimize the context-based reward estimator.
> >
> > **Regarding  W4:**
> >
> > We conduct these experiments to illustrate why the context-based reward estimator is chosen to measure how effectively the context captures task characteristic information. Specifically, neither the individual use of the context-based dynamic or reward estimator nor their combined use yields significant performance improvement. As shown in Figure 9 and Appendix G.3, setting an appropriate negative target is crucial. However, constructing a negative target for each next state is challenging due to its high dimensionality, requiring a multidimensional Gaussian distribution of noise to generate corresponding {{s'}_{i}^{t+1}}^{neg}. Our additional results show that dynamic estimations improve performance slightly in the Half-Cheetah-Vel environment when combined with reward estimations, while TCMRL with dynamic estimations alone shows the lowest performance in most environments. In contrast, constructing {{r'}_{i}^{t}}^{neg} is simpler due to its lower dimensionality and does not require such complexity. There are other methods for constructing negative targets and measuring context effectiveness, which we plan to explore in future work.
> >
> > [1] CLIMS: Cross Language Image Matching for Weakly Supervised Semantic Segmentation. 2022.
> >
> > [2] Weakly Supervised Action Localization by Sparse Temporal Pooling Network. 2018.
> >
> > [3] Self-Erasing Network for Integral Object Attention. 2018.
> >
> > [4] Salient Object Detection via Integrity Learning. 2023.

---

### Official Review · Reviewer_4bUz · 2024-11-04

**Soundness:** 3
**Presentation:** 3
**Contribution:** 3
**Rating:** 8
**Confidence:** 3

**Summary:**

The paper proposes a framework called Task Characteristic and Contrastive Contexts for Offline Meta-Reinforcement Learning (TCMRL), which aims to enhance the generalization ability of context-based offline meta-RL methods. TCMRL introduces two key components: a task characteristic extractor and a task contrastive loss, which work together to generate more comprehensive contexts by capturing both characteristic and contrastive task information. The task characteristic extractor uses an attention mechanism to emphasize transitions that are crucial for characterizing a task, while the task contrastive loss helps to distinguish different tasks by exploring interrelations among trajectory subsequences. Experiments demonstrate that TCMRL significantly improves adaptation to unseen tasks, outperforming existing offline meta-RL methods on multiple benchmark datasets.

**Strengths:**

1. TCMRL brings a fresh perspective by dynamically disentangling characteristic features from the trajectories while also maximizing interrelations among tasks.

2. The paper is written clearly, with a logical structure that makes it easy for readers to follow the flow of ideas. Key concepts such as task characteristic and contrastive information are well explained, with visual aids like figures and pseudocode to help illustrate the framework.

3. TCMRL improves the generalization capability of context-based offline meta-RL.

**Weaknesses:**

1. Some results are reported as normalized scores, e.g. Table 1. However, there is no explanation for how normalization is processed.

2. Although context shift is highlighted as one of the primary issues that TCMRL aims to solve, there is no in-depth analysis of how TCMRL reduces context shift compared to other methods, and potential limitations.

**Questions:**

See above.

---

> ### Author Response · Authors · 2024-11-18
> **Response to weaknesses**
>
> Thank you for taking the time to review our paper and for providing valuable feedback.
>
> ``W1. Some results are reported as normalized scores, e.g. Table 1. However, there is no explanation for how normalization is processed.``
>
> As shown in Figure 4, the performance of TCMRL and all baselines has largely converged within the training steps we set, under the conditions defined in Line 430, Section 5.2.
> For a specific task, we compute the average reward obtained from the steps after performance convergence. We then divide this average reward by its corresponding average return of the offline dataset to calculate the normalized score. The average returns of the offline datasets across all experimental environments are presented in Appendix H.
>
> ``W2. Although context shift is highlighted as one of the primary issues that TCMRL aims to solve, there is no in-depth analysis of how TCMRL reduces context shift compared to other methods, and potential limitations.``
>
> TCMRL is supported by both empirical evidence and existing theoretical foundations. We have introduced the concept of context shift (see Lines 39-45, Section 1), the limitations of existing context-based offline meta-RL methods (FOCAL (Li et al., 2021b), FOCAL++ (Li et al., 2021a), CORRO (Yuan & Lu, 2022), IDAQ (Wang et al., 2023) and CSRO (Gao et al., 2023)) (see Lines 136-144, Section 2), the motivation of the task characteristic extractor (see Lines 207-217, Section 4.1.1) and the motivation of the task contrastive loss (Lines 307-317, Section 4.1.2).
>
> Building on these statements already presented in our paper, we will further analyze TCMRL. An important reason why existing methods are significantly impacted by context shift is that they do not consider constructing comprehensive internal relationships among tasks from the offline data of meta-training tasks. In this context, TCMRL aims to reduce the Impact of context shift by learning how to effectively extract task information from offline data and constructing these internal relationships. These relationships are directly reflected in the contexts of the same task and the contexts of different tasks. Our goal is to capture task characteristic information and task contrastive information to generate contexts that contain both consistency within the same task and distinctness among different tasks. This allows us to construct comprehensive internal relationships among tasks, enabling the knowledge from offline datasets of meta-training tasks to be extended to unseen target tasks based on these relationships.
>
> **Potential limitation:**
>
> As demonstrated in Appendix G.4, when constructing the task contrastive loss to capture task contrastive information, we use subtrajectories with a length determined by a hyperparameter K. It means that we may require additional time to tune this hyperparameter when facing a new environment. In future work, we plan to explore more effective ways to design the task contrastive loss to mitigate this limitation.

---

> > ### Comment · Reviewer_4bUz · 2024-11-27
> >
> > Thank you for the explanations, which address my concerns. I'm happy to keep the current rating.

---

> > > ### Author Response · Authors · 2024-11-27
> > > **Official Comment by Authors**
> > >
> > > Thank you very much, we deeply appreciate the effort and time you've dedicated to providing us with your valuable feedback!

---

### Meta-Review · Area_Chair_JFnV · 2024-12-20

**Metareview:**

**summary**

The paper introduces TCMRL, a framework designed to enhance generalization and adaptability in context-based offline meta-RL methods. TCMRL incorporates a task characteristic extractor, which uses attention mechanisms to identify key transitions within tasks, and a task contrastive loss, which employs contrastive learning to differentiate tasks by analyzing interrelations among trajectory subsequences. Together, these components create comprehensive context representations that capture both task-specific and distinguishing features, enabling rapid adaptation to unseen tasks. The authors also condition a reward network on weighted transition encodings and penalizes reversed context weights to further refine task understanding. Experiments across benchmark datasets demonstrate that TCMRL outperforms existing approaches.

**strengths**

The paper is well-written with a clear and logical structure, making it easy to follow, and it demonstrates consistent improvements over baseline methods across a diverse set of experiments.


**weaknesses**

* The paper lacks a clear technical rationale for its design choices, making it unclear why these specific components achieve the desired behavior of the context encoder.
* There is no theoretical analysis or strong empirical justification to demonstrate that the proposed objective function guarantees generalization
* The approach combines modified versions of existing loss functions and established techniques, limiting the novelty of the contribution.

**decision**

It is difficult to say that the contributions of this work are very significant due to several limitations listed in weaknesses. I recommend that the authors address these concerns and consider resubmitting to another venue.

**Additional Comments On Reviewer Discussion:**

I don’t think that the authors made a valid argument to address the reviewer's concerns about novelty.

---

### Decision · Program_Chairs · 2025-01-22

Reject